# Minimal Value-Equivalent Partial Models for Scalable and Robust Planning in Lifelong Reinforcement Learning

## Abstract

Learning models of the environment from pure interaction is often considered an essential component of building lifelong reinforcement learning agents. However, the common practice in model-based reinforcement learning is to learn models that model every aspect of the agent's environment, regardless of whether they are important in coming up with optimal decxisions or not. In this paper, we argue that such models are not particularly well-suited for performing scalable and robust planning in lifelong reinforcement learning scenarios and we propose new kinds of models that only model the relevant aspects of the environment, which we call *minimal value-equivalent partial models*. After providing the formal definitions of these models, we provide theoretical results demonstrating the scalability advantages of performing planning with such models and then perform experiments to empirically illustrate our theoretical results. Finally, we provide some useful heuristics on how to learn these kinds of models with deep learning architectures and empirically demonstrate that models learned in such a way can allow for performing planning that is robust to distribution shifts and compounding model errors. Overall, both our theoretical and empirical results suggest that minimal value-equivalent partial models can provide significant benefits to performing scalable and robust planning in lifelong reinforcement learning scenarios.

## 1 Introduction

It has long been argued that in order for reinforcement learning (RL) agents to perform well in lifelong RL (LRL) scenarios, they should be able to learn a model of their environment, which allows for advanced computational abilities such as counterfactual reasoning and fast re-planning (Sutton & Barto, 2018; Schaul et al., 2018; Sutton et al., 2022). Even though this is a widely accepted view in the RL community, the question of what *kinds* of models would better suite for performing LRL still remains unanswered. As LRL scenarios involve large environments with lots of irrelevant aspects and periodic or non-periodic distribution shifts, directly applying the ideas developed in the classical model-based RL literature (see e.g., Ch. 8 of Sutton & Barto, 2018) to these problems is likely to lead to catastrophic results in building scalable and robust lifelong learning agents. Thus, there is a need to rethink some of the ideas developed in the classical model-based RL literature while developing new concepts and algorithms for performing model-based RL in LRL scenarios.

In this paper, we argue that one important idea to reconsider is whether if the agent's model should model every aspect of its environment. In classical model-based RL, the learned model is a model over every aspect of the environment. However, due to the large state spaces of LRL environments, these types of models are likely to lead to serious problems in performing scalable model-based RL, i.e., in quickly learning a model and in quickly performing planning with the learned model to come up with an optimal policy. Also, due to the inherent non-stationarity of LRL environments, these types of detailed models are likely to lead to models that overfit to the irrelevant aspects of the environment and cause serious problems in performing robust model-based RL, i.e., learning & planning with models that are robust to distributions shifts and compounding model errors.

To this end, we argue that models that *only* model the relevant aspects of the agent's environment, which we call *minimal value-equivalent partial models*, would be better suited for performing

model-based RL in LRL scenarios. We first start by developing the theoretical underpinnings of how such models could be defined and studied in model-based RL. Then, we provide theoretical results demonstrating the scalability advantages, i.e., the value and planning loss and computational and sample complexity advantages, of performing planning with minimal value-equivalent partial models and then perform several experiments to empirically illustrate these theoretical results. Finally, we provide some useful heuristics on how to learn these kinds models with deep learning architectures and empirically demonstrate that models learned in such a way can allow for performing planning that is robust to distribution shifts and compounding model errors. Overall, both our theoretical and empirical results suggest that minimal value-equivalent partial models can provide significant benefits to performing scalable and robust model-based RL in LRL scenarios. We hope that our study will bring the community a step closer in building model-based RL agents that are able to perform well in LRL scenarios.

## 2 BACKGROUND

**Reinforcement Learning.** In RL (Sutton & Barto, 2018), an agent interacts with its environment through a sequence of actions to maximize its long-term cumulative reward. Here, the environment is usually described as a Markov decision process (MDP) $M \equiv (\mathcal{S}, \mathcal{A}, P, R, \gamma)$, where $\mathcal{S}$ and $\mathcal{A}$ are the (finite) set of states and actions, $P : \mathcal{S} \times \mathcal{A} \times \mathcal{S} \to [0, 1]$ is the transition distribution, $R : \mathcal{S} \times \mathcal{A} \to [0, R_{\max}]$ is the reward function, and $\gamma \in [0, 1)$ is the discount factor. On the agent's side, through the use of a perfect state encoder $\phi^* : \mathcal{S} \to \mathcal{F}$, every state $s \in \mathcal{S}$ can be represented, without any loss of information, as an $n$-dimensional feature vector $f = [f_1, f_2, \ldots, f_n]^\top \in \mathcal{F}$, which consists of $n$ different features $\mathbb{F} = \{f_i\}_{i=1}^n$ where $f_i \in \mathcal{F}_i \ \forall i \in \{1, \ldots, n\}$ (also see Boutilier et al. (2000)). Note that as there is no loss of information, $\mathbb{F}$ contains all the possible features that are relevant in describing the states of the environment. Thus, from the agent's side, the MDP $M$ can losslessly be represented as another MDP $m^* = (\mathcal{F}, \mathcal{A}, p^*, r^*, \gamma)$, where $\mathcal{F}$ and $\mathcal{A}$ are the (finite) set of feature vectors and actions, $p^* : \mathcal{F} \times \mathcal{A} \times \mathcal{F} \to [0, 1]$ and $r^* : \mathcal{F} \times \mathcal{A} \to [0, R_{\max}]$ are the transition distribution and reward function, and $\gamma \in [0, 1)$ is the discount factor. For convenience, we take the agent's view and refer to the environment as $m^*$ throughout this study. The goal of the agent is to learn a value estimator $Q : \mathcal{F} \times \mathcal{A} \to \mathbb{R}$ that induces a policy $\pi \in \Pi \equiv \{\pi \mid \pi : \mathcal{F} \times \mathcal{A} \to [0, 1]\}$, maximizing $E_{\pi, p^*}[\sum_{t=0}^\infty \gamma^t r^*(F_t, A_t) \mid F_0]$ for all $F_0 \in \mathcal{F}$.

**Model-Based RL.** One of the prevalent ways of achieving this goal is through the use of model-based RL methods in which there are two main phases: the learning and planning phases. In the learning phase, the gathered experience is mainly used in learning an encoder $\phi : \mathcal{S} \to \mathcal{F}$ and a model $m \equiv (p, r) \in \mathcal{M} \equiv \{(p, r) \mid p : \mathcal{F} \times \mathcal{A} \times \mathcal{F} \to [0, 1], r : \mathcal{F} \times \mathcal{A} \to [0, R_{\max}]\}$, and optionally, the experience may also be used in improving the value estimator. In the planning phase, the learned model $m$ is then used either for solving for the fixed point of a system of Bellman equations (Bellman, 1957), or for simulating experience, either to be used alongside real experience in improving the value estimator, or just to be used in selecting actions at decision time (Alver & Precup, 2022; Sutton & Barto, 2018).

**Value-Equivalence.** One of the recent trends in model-based RL is to learn models that are specifically useful for value-based planning (see e.g., Silver et al., 2017; Schrittwieser et al., 2020), which has been recently formalized in several different ways through the studies of Grimm et al. (2020; 2021). Inspired by these studies, we define a related form of value-equivalence as follows. Let $V_m^\pi \in \mathbb{R}^{|\mathcal{F}|}$ be the value vector of a policy $\pi \in \Pi$ evaluated in model $m$, whose elements are defined $\forall f \in \mathcal{F}$ as $V_m^\pi(f) \equiv E_{\pi, p}[\sum_{t=0}^\infty \gamma^t r(F_t, A_t) | F_0 = f]$, and let $V_m^* \in \mathbb{R}^{|\mathcal{F}|}$ be the optimal value vector in model $m$. We say that a model $m \in \mathcal{M}$ is a *value-equivalent (VE) model* of the true environment $m^* \in \mathcal{M}$ if the following equality holds:

$$V_{m^*}^{\pi_m^*} = V_{m^*}^* \quad \forall \pi_m^* \in \Pi, \tag{1}$$

where $\pi_m^*$ is an optimal policy obtained as a result of planning with model $m$.

## 3 MINIMAL VALUE-EQUIVALENT PARTIAL MODELS

In classical model-based RL (Ch. 8 of Sutton & Barto, 2018), an agent learns a very detailed model of its environment that models every aspect of it, regardless of whether these aspects are relevant

in the process of coming up with optimal decisions or not. However, in LRL scenarios, where the agent is "small" and the environment is "vast" (Schaul et al., 2018), this approach is likely to be problematic as modeling every aspect of the environment becomes quite impractical. Even if the agent overcomes its capacity limitations and manages to model every aspect, as we will demonstrate, these kinds of detailed models can lead to large planning losses and dramatically slowdown both the model-learning and planning processes. And, as we will further demonstrate, detailed models can also be fragile to the distribution shifts in the environment and to the compounding model errors that happen during the unrollment of the learned model. In order to overcome these challenges, we start by proposing new kinds of models that only model certain aspects, either relevant or irrelevant, of the agent's environment. For this, we first start by clarifying the notion of "aspect": in this study, by "aspect", we mean a feature of the environment $f_i \in \mathcal{F}_i$ that is learnable by the agent (see Sec. 2). We are now ready to define *partial models*:

**Definition 1** (Partial Models). *Given a set of features $\mathbb{F}$, let $\mathbb{F}_P \subset \mathbb{F}$ s.t. $|\mathbb{F}_P| < |\mathbb{F}|$. Let $\mathcal{F}_P$ be a space of feature vectors in which the feature vectors consist the features in $\mathbb{F}_P$. We say that a model $m_P$ is a* partial model *of the true environment $m^* \in \mathcal{M}$ if it is defined over the feature vector space $\mathcal{F}_P$, i.e., $m_P \in \mathcal{M}_P \equiv \{(p_P, r_P) \mid p_P : \mathcal{F}_P \times \mathcal{A} \times \mathcal{F}_P \to [0,1], r_P : \mathcal{F}_P \times \mathcal{A} \to [0, R_{\max}]\}$.*

According to Defn. 1, any model that only models certain features of the environment is a partial model of the environment $m^* \in \mathcal{M}$. However, in order for a partial model to be useful, it should be able to model the relevant features of the environment that allow for achieving the task of interest. In order to separate out the relevant features from the irrelevant ones, we define the relevant ones as:

**Definition 2** (Relevant Features). *Given a set of features $\mathbb{F}$, let $\mathbb{F}_R \subset \mathbb{F}$. Let $\mathcal{F}_R$ be a space of feature vectors in which the feature vectors consist of the features in $\mathbb{F}_R$. We say that the features $f_i \in \mathbb{F}_R$ are* relevant features *of the task of interest if they are necessary and sufficient for defining a space of models $\mathcal{M}_R \equiv \{(p_R, r_R) \mid p_R : \mathcal{F}_R \times \mathcal{A} \times \mathcal{F}_R \to [0,1], r_R : \mathcal{F}_R \times \mathcal{A} \to [0, R_{\max}]\}$ that contains value-equivalent models of the true environment $m^* \in \mathcal{M}$.*

Now that we have defined partial models and distinguished between the relevant and irrelevant features of the environment, we are ready to define an important class of partial models that at the very least model the relevant aspects of the environment:

**Definition 3** (VE Partial Models). *Given a set of features $\mathbb{F}$, let $\mathbb{F}_{VEP} \subset \mathbb{F}$ s.t. $|\mathbb{F}_{VEP}| < |\mathbb{F}|$ and $\mathbb{F}_R \subseteq \mathbb{F}_{VEP}$. Let $\mathcal{F}_{VEP}$ be a space of feature vectors in which the feature vectors consist of the features in $\mathbb{F}_{VEP}$. Let $m_{VEP}$ be a partial model that is defined over the feature vector space $\mathcal{F}_{VEP}$, i.e., $m_{VEP} \in \mathcal{M}_{VEP} \equiv \{(p_{VEP}, r_{VEP}) \mid p_{VEP} : \mathcal{F}_{VEP} \times \mathcal{A} \times \mathcal{F}_{VEP} \to [0,1], r_{VEP} : \mathcal{F}_{VEP} \times \mathcal{A} \to [0, R_{\max}]\}$. We say that $m_{VEP}$ is a* VE partial model *of the true environment $m^* \in \mathcal{M}$ if it is a VE model of $m^*$, i.e.,*

$$V_{m^*}^{\pi^*_{m_{VEP}}} = V_{m^*}^* \quad \forall \pi^*_{m_{VEP}} \in \Pi_{VEP}, \tag{2}$$

*where $\pi^*_{m_{VEP}}$ is an optimal policy obtained as a result of planning with model $m_{VEP}$ and $\Pi_{VEP} \equiv \{\pi \mid \pi : \mathcal{F}_{VEP} \times \mathcal{A} \to [0,1]\}$.*

Although it is important to learn partial models that at the very least model the relevant aspects of the environment, as we will theoretically and empirically demonstrate, partial models are mostly beneficial when they *only* model the relevant aspects of the environment, i.e., when $\mathbb{F}_{VEP} = \mathbb{F}_R$. We refer to these models as **minimal VE partial models**. Note that minimal VE partial models are a special class of VE partial models, and VE partial models are a special class of partial models.

**Illustrative Example.** As an illustration of the models defined above, let us start by considering the Squirrel's World (SW) environment depicted in Fig. 1, in which the squirrel's (the agent) job is to navigate from cell E1 to cell E16 to pickup the nut without getting caught by the hawk that flies back and forth horizontally along row C. At each time step, the squirrel receives

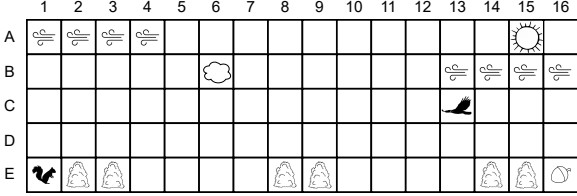

Figure 1: The SW environment.

as input an $5 \times 16$ image of the current state of the environment and then, through the use of a pre-defined state encoder, transforms this image into a feature vector that contains information regarding all aspects of the current state of the environment, i.e., the feature vector contains information on the

current position of the squirrel, hawk and the cloud, the current direction of the hawk, the current wind direction in rows A and B and the current weather condition. Based on this, the squirrel selects an action that either moves it to the left or right cell, or keeps it position fixed. If the squirrel gets caught by the hawk or if it is out of time, it receives a reward of $0$ and the episode terminates, and if the squirrel successfully navigates to the nut, it gets a reward of $+10$ and the episode terminates. In this environment, as the hawk moves 5x the speed of the squirrel, a straightforward policy of always moving to the right will not get the squirrel to the nut. Thus, the squirrel has to come up with non-trivial policies that take into account both the cells with bushes (see e.g., cells E2, E3), which allow for sheltering, and the position and direction of the hawk.

In this environment, examples of partial models can be a model that only models the cloud position and the wind direction for rows A and B, or a model that only models the weather condition and the hawk's direction. However, for a partial model to be VE or minimal VE, it has to model the relevant features for the tasks of interest which is reaching the nut. In the SW environment, there are three relevant features: (i) the squirrel's position, (ii) the hawk's position, and (iii) the hawk's direction, as the squirrel would have to have access to all three of these features to come up with optimal policies. Thus, an example of a VE partial model can be a model that models both the three relevant features and the weather condition, and an example of a minimal VE partial model can be a model that only models the three relevant features.

## 4 THEORETICAL RESULTS

In this section, we first analyze the value and planning losses (Sec. 4.1) of VE partial models and then derive formal results demonstrating the computational and sample complexity benefits (Sec. 4.2) of using such models. We then discuss scenarios where the VE partial model is a minimal one.

### 4.1 VALUE AND PLANNING LOSS ANALYSES

We start our formal analysis by studying the value loss incurred due to planning with a VE partial model $m_{VEP}$ in place of the true environment $m^*$. To simplify the analysis, we assume that the agent already has access to this model and does not need to learn it.

**Theorem 1.** *Let $m_{VEP} \in \mathcal{M}_{VEP}$ be a VE partial model of the true environment $m^* \in \mathcal{M}$. Then, the value loss between an optimal policy in $m^*$, $\pi^*$, and an optimal policy in $m_{VEP}$, $\pi^*_{m_{VEP}}$ is given by:*

$$\left\| V^*_{m^*} - V^{\pi^*_{m_{VEP}}}_{m^*} \right\|_\infty = 0. \tag{3}$$

Due to space constraints, we defer all the proofs to App. A. Theorem 1 says that by planning with a (non-minimal or minimal) VE partial model, an agent would incur no value loss compared to planning with the true environment itself.

Next, we study the planning loss (Jiang et al., 2015) incurred due to planning with an approximate VE partial model $\tilde{m}_{VEP} \in \mathcal{M}_{VEP}$ in place of the actual VE partial model $m_{VEP} \in \mathcal{M}_{VEP}$. Similar to Jiang et al. (2015), we also consider the certainty-equivalence control setting in which the agent acts according to a policy that is optimal with respect to its current approximate model.

**Theorem 2.** *Let $m_{VEP} \in \mathcal{M}_{VEP}$ be a VE partial model of the true environment $m^* \in \mathcal{M}$, and let $\tilde{m}_{VEP} \in \mathcal{M}_{VEP}$ be model that comprises of the reward function of $m_{VEP}$ and a transition distribution that is estimated from $n$ samples for each $(f, a)$ pair. Let $\Pi_{r_{VEP}} \equiv \{\pi \mid \exists\, p_{VEP} \text{ s.t } \pi \text{ is optimal in } (p_{VEP}, r_{VEP})\}$. Then, certainty-equivalence planning with $\tilde{m}_{VEP}$ has planning loss:*

$$\left\| V^*_{m_{VEP}} - V^{\pi^*_{\tilde{m}_{VEP}}}_{m_{VEP}} \right\|_\infty \leq \frac{2R_{\max}}{(1-\gamma)^2} \sqrt{\frac{1}{2n} \log \frac{2|\mathcal{F}_{VEP}||\mathcal{A}||\Pi_{r_{VEP}}|}{\delta}}, \tag{4}$$

*with probability at least $1 - \delta$.*

Theorem 2 implies that given a fixed amount of data, the upper bound of the planning loss of a VE partial model depends on both the size of its feature vector space, $|\mathcal{F}_{VEP}|$, and the size of its policy class being searched over by planning, $|\Pi_{r_{VEP}}|$.[1] This in turn implies that, given a fixed amount of

---

[1]Note that $|\mathcal{F}_{VEP}|$ also affects $|\Pi_{r_{VEP}}|$, i.e., as $|\mathcal{F}_{VEP}|$ grows, $|\Pi_{r_{VEP}}|$ also grows.

data, compared to a regular model, a VE partial model is likely to have less planning loss and this loss is likely to be minimized when the VE partial model is a minimal one.

## 4.2 COMPUTATIONAL AND SAMPLE COMPLEXITY BENEFITS

We now study the computational and sample complexity benefits of performing model-based RL with VE partial models. Due to the well-established theoretical results around it, we choose to study these benefits in the context of value iteration (Bertsekas & Tsitsiklis, 1996). However, we note that the implications of our results would apply to a wide variety of planning algorithms.

Starting with the computational complexity benefits, it is well-known that the computational complexity of performing a single step of value iteration with an arbitrary model $m \in \mathcal{M}$ is $\mathcal{O}(|\mathcal{F}|^2|\mathcal{A}|)$ (Agarwal et al., 2022). Thus, the computational complexity of performing a single step of value iteration with a VE partial model $m_{VEP} \in \mathcal{M}_{VEP}$ would be $\mathcal{O}(|\mathcal{F}_{VEP}|^2|\mathcal{A}|)$. This implies that compared to planning with regular models, planning with VE partial models would provide a significant computational complexity benefit and this benefit would be maximized when the model used for planning is a minimal VE partial model.

Moving on to the sample complexity benefits, previous studies of Kearns & Singh (1998); Kakade (2003); Azar et al. (2012) have shown that the sample complexity of obtaining an $\varepsilon$ estimation of the optimal action value function through the use of Q-value iteration (see Alg. 1) given access only to a generative model is in the order of the magnitude of the model's state and action space. Building on top of this result, we now study the sample complexity benefits of planning with approximate VE partial models that are obtained as a result of sampling generative VE partial models.

**Theorem 3.** *Let $m_{VEP} \in \mathcal{M}_{VEP}$ be a VE partial model of the true environment $m^* \in \mathcal{M}$. Let $\tilde{m}_{VEP} \in \mathcal{M}_{VEP}$ be the corresponding approximate VE partial model that has the same reward function as $m_{VEP}$, but whose transition distribution is estimated by $m$ calls to the generative model $m_{VEP}$, where*

$$m = \mathcal{O}\left(\frac{|\mathcal{F}_{VEP}||\mathcal{A}|}{(1-\gamma)^4 \varepsilon^2}\right), \tag{5}$$

*and let $Q_{\tilde{m}_{VEP}}^k$ be the value returned by Q-value iteration at the kth epoch. Then, with probability greater than $1 - \delta$, the following holds for all $f \in \mathcal{F}_{VEP}$ and $a \in \mathcal{A}$:*

$$\left\|Q_{\tilde{m}_{VEP}}^k - Q_{m_{VEP}}^*\right\|_\infty \le \varepsilon, \tag{6}$$

*where $k = \frac{\log(\varepsilon(1-\gamma))}{\log \gamma}$ and $Q_{m_{VEP}}^*$ is the optimal action value function in $m_{VEP}$.*

Theorem 3 implies that compared to a regular model, a VE partial model is likely to require less samples in obtaining an $\varepsilon$ estimation of the optimal action value function through the use of Q-value iteration with a generative model, and the number of samples required is likely to be minimized when the VE partial model is a minimal one.

## 5 EXPERIMENTAL RESULTS

We start this section by performing experiments to demonstrate the scalability advantages of minimal VE partial models, which are illustrations of the theoretical results derived in Sec. 4, and then we perform experiments to demonstrate the robustness advantages of these models. The details of our experiments can be found in App. C.

**Environments.** We perform experiments on both the SW environment (see Fig. 1) and on variations of the Two Rooms Dynamic Obstacles (2RDO) environment that are built on top of Minigrid (Chevalier-Boisvert et al., 2018) (see Fig. 2), as these environments allow for designing controlled experiments that are helpful in answering the questions of interest to this study. Some of the details of the SW environment are already presented in Sec. 3 and we refer the reader to App. C for more details. In the 2RDO environments, the agent, depicted by the red triangle, spawns in top-left of the top room and has to navigate to the green goal cell located in the bottom-right of the same room, regardless of the gaseous motions of the obstacles in the bottom room. At each time step, the agent receives an image of the current state of the grid and then, through the use of a learned state encoder, transforms this image into a feature vector. Based on this, the agent selects an action that either turns

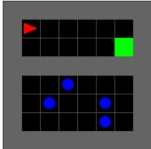 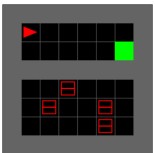 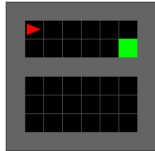 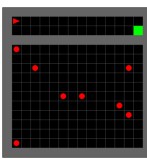 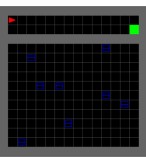

   (a) 8x8 BlueBalls    (b) 8x8 RedBoxes   (c) 8x8 NoObstacles  (d) 16x16 RedBalls  (e) 16x16 BlueBoxes

Figure 2: Variations of the 2RDO environment with grid sizes of 8x8 and 16x16. In these environments, there are either no obstacles (c), or there are several obstacles (balls and boxes) with different colors (a, b, d, e).

it left or right, or moves it forward. If the agent successfully navigates to the goal cell, it receives a reward of $+1$ and the episode terminates. More details on the 2DRO environments can be found in App. C as well.

## 5.1 SCALABILITY EXPERIMENTS

For our scalability experiments, we perform experiments with several non-VE ($m_1$, $m_2$, $m_3$) and VE ($m_4$, $m_5$, $m_6$) partial models of both the deterministic and stochastic versions of the SW environment, referred to as Det-SW and Stoch-SW, respectively. The details of these models can be found in Table 1. For all of our experiments, we use value iteration as our planning algorithm.

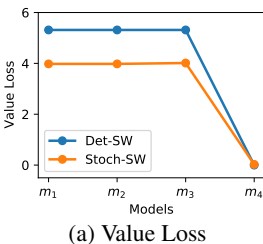 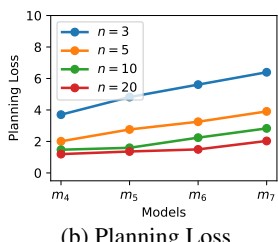 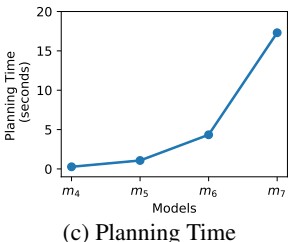

       (a) Value Loss             (b) Planning Loss          (c) Planning Time

Figure 3: The (a) value losses, (b) planning losses, and (c) planning times of several models. Plot (a) was obtained over a single run and plots (b) and (c) were obtained by averaging over 50 runs per model.

**Question 1.** *Do minimal VE partial models allow for planning with no value loss?*

In Sec. 4.1, we argued that by planning with a (non-minimal or minimal) VE partial model, an agent would incur no value loss compared to planning with the true environment itself. To empirically verify this, we present the agent with a set of non-VE partial models $m_1, m_2, m_3$ and a minimal VE partial model $m_4$, and compare the value losses on both the Det-SW and Stoch-SW environments. Results are shown in Fig. 3a. We can indeed see that while the VE partial model incurs no value loss, the non-VE ones do incur serious value losses.

**Question 2.** *Do minimal VE partial models allow for planning with less planning loss?*

In Sec. 4.1, we argued that given a fixed amount of data, compared to a regular model, a VE partial model is likely to incur less planning loss, and this loss is likely to be minimized when the VE partial model is a minimal one. For empirical verification, we compare the planning losses of a minimal VE partial model $m_4$, two (non-minimal) VE partial models $m_5$ and $m_6$, and a regular model $m_7$, across dataset sizes of 3, 5, 10 and 20, which corresponds to the number of samples for each $(f, a)$ pair, on the Stoch-SW environment. Results in Fig. 3b show that, as expected, VE partial models indeed incur less planning losses than regular models, and the minimal VE partial model incurs the least planning loss.

**Question 3.** *Do minimal VE partial models provide computational complexity benefits?*

In Sec. 4.2, we argued that compared to regular models, planning with VE partial models would provide a significant computational complexity benefit and this benefit would be maximized when the model used for planning is a minimal VE partial model. To empirically verify this, we present the agent with a minimal VE partial model $m_4$, two VE partial models $m_5$ and $m_6$, and a regular model $m_7$ of the Det-SW environment, and compare the average time it takes to perform a single step of value iteration for each of these models. Results are shown in Fig. 3c. As can be seen, planning with VE partial models indeed provides significant computational complexity benefits, and this benefit is maximized when the VE partial model is a minimal one.

**Question 4.** *Do minimal VE partial models provide sample complexity benefits?*

In Sec. 4.2, we argued that compared to regular models, planning with VE partial models is likely to provide a sample complexity benefit and this benefit is likely to be maximized when the model that is used for planning is a minimal VE partial model. For empirical verification, we present the agent with a minimal VE partial model $m_4$ and with a regular model $m_7$ as generative models, and compare the sample efficiencies, as a result of performing Q-value iteration, on the Det-SW and Stoch-SW environments. In these experiments, after every episodic interaction, the agent updates its

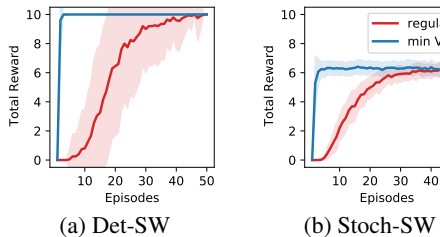

(a) Det-SW     (b) Stoch-SW

Figure 4: The total reward obtained as a result of planning with models $m_4$ and $m_7$ on the (a) Det-SW and (b) Stoch-SW environments. Shaded regions are standard errors over 50 runs.

model with the collected trajectory, and then performs Q-value iteration until convergence. Results in Fig. 4 show that, as expected, planning with minimal VE partial models indeed provides significant sample efficiency benefits compared to planning with regular models.

## 5.2 ROBUSTNESS EXPERIMENTS

For our robustness experiments, we perform experiments on variations of the 2RDO environment with grid sizes of 8x8 and 16x16. For convenience, we will refer to these environments with their grid size followed by their obstacle type. For example. we will refer to the 8x8 2DRO environment with red balls as 8x8 RedBalls (see Fig. 2). For all of our experiments, we use the straightforward decision-time planning algorithm of Zhao et al. (2021) (see Alg. 2) whose details can be found in App. C. As this algorithm makes use of neural networks, before moving on to the robustness experiments, we try to answer the following question.

**Question 5.** *How to learn minimal VE partial models with deep learning architectures?*

So far, for illustration purposes, we have only performed experiments in which we had a direct control over the features of the agent's model (see the models in Table 1). However, in realistic scenarios, the agent would have to come up on its own with a set of features to build a model of the only relevant aspects of its environment. A very popular way of letting the agent come up with its own features is to use neural networks in the representation of the agent's encoder, value estimator and model, and then to train it end-to-end on the environment of interest. However, in order for the agent to come up with only the relevant features, it has to be trained with the right inductive biases. Even though finding the right inductive biases to train a model-free or model-based RL agent is still an open problem in the representation learning literature (Bengio et al., 2013), in this study, we propose two inductive biases that are likely to guide the agent in coming up with only the relevant features. The first one is to only let the value estimator shape the encoder and prevent the model from doing so (see Fig. 7). In this way, the agent can be guided in learning the features that are relevant for predicting the right values in the environment. And, the second one is to train the agent across a variety of environments in which the irrelevant aspects keep changing and the relevant ones stay the same. In this way, the agent can be guided in not learning the irrelevant aspects of the environment.

In order to test the usefulness of these two inductive biases in coming up with only the relevant features of the environment, we compare three different agents: (i) a regular agent, $A_{REG}$, that was trained on the 8x8 BlueBalls environment and whose encoder was jointly shaped by its value estimator and model, (ii) an agent, $A_{VES}$, that was again trained on the 8x8 BlueBalls environment, but whose encoder was only shaped by its value estimator, and (iii) an agent, $A_{VES+ME}$, that was trained on the 8x8 BlueBalls, GreenBalls, PurpleBalls and YellowBalls environments and whose encoder was only shaped by its value estimator. We compare these agents on the 8x8 BlueBalls and NoObstacles environments. If the agent is successful in coming up with only the relevant features of the environment, which are the positions of the agent and the goal, and not the positions and motions of the obstacles, we would expect it to perform similarly on the 8x8 BlueBalls and 8x8 NoObstacles environments. Results are shown in Fig. 5a & 5b. As can be seen, even though all of the agents perform well on the 8x8 BlueBalls environment, the $A_{REG}$ agent completely fails on the 8x8 NoObstacles environment, demonstrating that without the necessary inductive biases an agent is

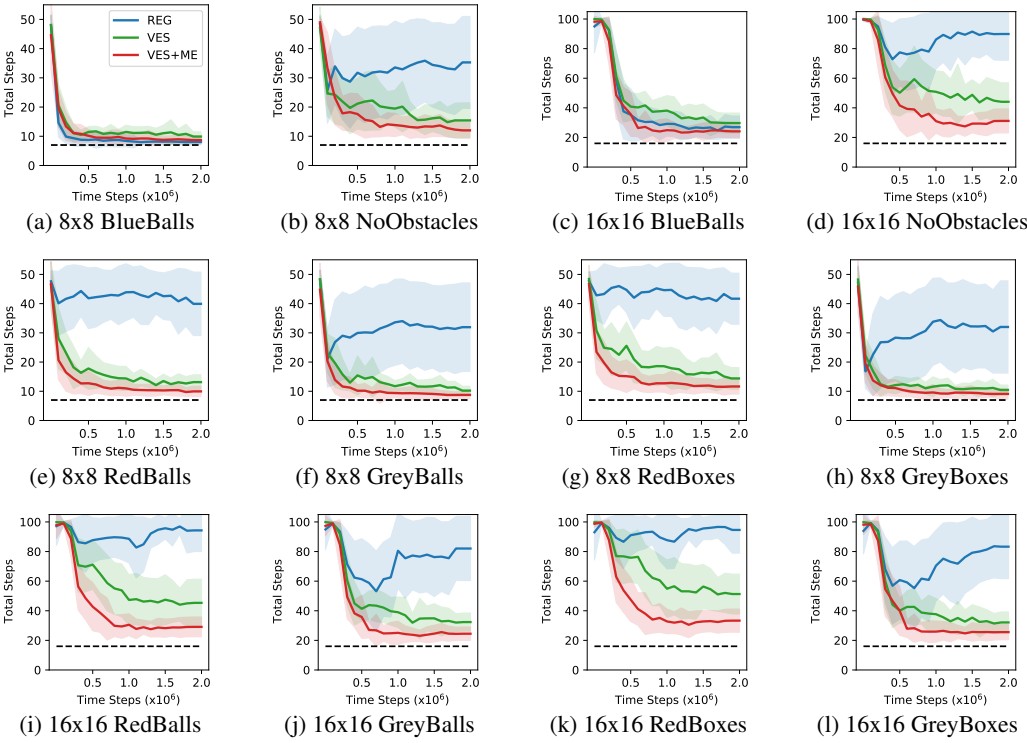

Figure 5: The total steps to reach the goal in the 8x8 and 16x16 versions of the (a, c) BlueBalls, (b, d) NoObstacles, (e, i) RedBalls, (f, j) GreyBalls, (g, k) RedBoxes and (h, l) GreyBoxes environments for the $A_{REG}$, $A_{VES}$ and $A_{VES+ME}$ agents. Black dashed lines indicate the performance of the optimal policy in the corresponding environments. Shaded regions are standard errors over 100 runs.

not capable of coming up with only the relevant features itself. We can also see that the $A_{VES}$ agent achieves a better performance than the $A_{REG}$ agent and that the $A_{VES+ME}$ agent achieves an even better performance than the $A_{VES}$ agent, demonstrating the usefulness of our proposed inductive biases in inducing models that display the behavior of minimal VE partial models. In order to test the scalability of our results, we have also performed the same experiments with 16x16 versions of the environments. As can be seen in Fig. 5c & 5d, we obtain similar results.

**Question 6.** *Can minimal VE partial models be useful for performing robust transfer?*

As minimal VE partial models only model the relevant aspects of the environment, we would expect them to be robust to the distribution shifts happening in the irrelevant aspects of the environment. In order to test this, we compare the performances of the $A_{REG}$, $A_{VES}$ and $A_{VES+ME}$ agents on the 8x8 and 16x16 RedBalls, GreyBalls, RedBoxes and GreyBoxes environments. Results are shown in Fig. 5e-5l. As can be seen, while the $A_{REG}$ agent fails and the $A_{VES}$ agent only shows signs of robust transfer, the $A_{VES+ME}$ agent is able to perform robust transfer without any problem. These results illustrate the ability of minimal VE partial models in performing robust transfer.

**Question 7.** *Are minimal VE partial models more robust to compounding model errors?*

As minimal VE partial models only model the relevant aspects of the environment, compared to regular models, we would expect them to be less susceptible to compounding model errors during planning. In order to test this, we compare the performances of the $A_{REG}$ and $A_{VES+ME}$ agents with search budgets of 20, 40 and 80 on the 16x16 BlueBalls environment. Note that this environment has been seen before by both of the agents. Results in Fig. 6 show that while the performance of $A_{REG}$ agent drops significantly with the increase in the search budget, the performance of the $A_{VES+ME}$ agent stays close to optimal, demonstrating the robustness of minimal VE partial models to compounding model errors.

## 6 RELATED WORK

**Partial Models.** In the context of RL, the initial studies of partial models can be dated back to the seminal study of Talvitie & Singh (2008) which proposes to learn several models of an uncontrolled dynamical systems that are partial at the observation level. In contrast, we propose to learn a single and useful partial model of a controlled dynamical system that is partial at the feature level, which provides several advantages such as eliminating the question of how to combine the learned models, using them for control purposes, and making them compatible with function approximation. Our work also has a very close connection to the study of Zhao et al. (2021) which proposes a transformer-based deep

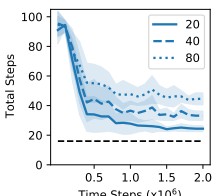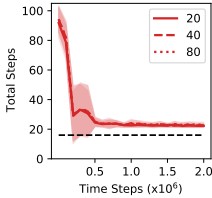

(a) The $A_{REG}$ agent  (b) The $A_{VES+ME}$ agent

Figure 6: The total steps to reach the goal in the 16x16 BlueBalls environment for the $A_{REG}$ and $A_{VES+ME}$ agents with search budgets of 20, 40 and 80. Black dashed lines indicate the performance of the optimal policy in the corresponding environments. Shaded regions are standard errors over 100 runs.

model-based agent that dynamically attends to relevant parts of its state representation during planning. However, our work differs in that we propose the general concept of partial models for LRL that is independent of the agent's implementation details. Lastly, another related line of research is the studies of Khetarpal et al. (2020; 2021) on affordances which focus on building models that partial in the action space. Our study is complementary to these studies in that they can still leverage (non-minimal or minimal) VE partial models to reduce the size of the feature space and further increase the benefits of performing model-based RL with partial models.

**Value-Equivalence.** A recent trend in model-based RL is to learn models that are specifically useful for value-based planning (see e.g. Silver et al., 2017; Oh et al., 2017; Farquhar et al., 2017; Schrittwieser et al., 2020; Grimm et al., 2020; 2021). Even though our work also advocates the idea that models should be useful in value-based planning, our work differs in that we also argue that the explicit partiality of the models can provide significant scalability and robustness benefits when performing model-based RL in LRL scenarios.

**Planning in Learned Feature Spaces.** Even though there has been recent studies that study the effect of the introduction of the irrelevant features in the agent's learned representation (Efroni et al., 2022a;b), our study differs in that we are mainly interested in LRL environments in which environment mostly consists of irrelevant features and the relevant features to the agent do not change over time. Our work is also different from the studies that learn models through self-supervised learning (see e.g., Sekar et al., 2020) in that we explicitly study the structure of the learned representation having relevant and irrelevant components.

## 7 CONCLUSION AND DISCUSSION

In conclusion, in this study, we have introduced special types of models, called minimal VE partial models, that only model the relevant aspects of the environment and are particularly useful in LRL scenarios. Our theoretical results suggest that these models can provide significant advantages in the value and planning losses that are incurred during planning and in the computational and sample complexity of planning. Our empirical results (i) validate our theoretical results and show that these models can scale to large environments, and (ii) show that these models can be robust to distribution shifts and compounding model errors. Overall, our findings suggest that minimal VE partial models can provide significant advantages in performing model-based RL in LRL scenarios. One limitation of our work is that, rather than providing a principled method, we have only provided several heuristics for training deep RL agents that can come up with only the relevant features of the environment. However, we note that this is mainly due to the lack of principled approaches in the representation learning literature, and we believe that this limitation can be overcomed with more principled approaches being introduced in the literature. We hope to tackle this limitation in future work. Another important limitation is that, due to the need to perform illustrative and controlled experiments, we have only performed experiments in the SW and 2RDO environments where there is just a single task and there is no sequence of tasks, requiring a model the same relevant features, that unfold over time. However, experiments in more environments that have this sequential nature can be helpful in further validating the advantages of minimal VE partial models in LRL scenarios, which we also hope to tackle in future work.

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

## A    PROOFS

**Theorem 1.** *Let $m_{VEP} \in \mathcal{M}_{VEP}$ be a VE partial model of the true environment $m^* \in \mathcal{M}$. Then, the value loss between an optimal policy in $m^*$, $\pi^*$, and an optimal policy in $m_{VEP}$, $\pi^*_{m_{VEP}}$ is given by:*

$$\left\| V^*_{m^*} - V^{\pi^*_{m_{VEP}}}_{m^*} \right\|_\infty = 0. \tag{7}$$

*Proof.* This result directly follows from Defn. 3. Recall that, according to Defn. 3, we have:

$$V^{\pi^*_{m_{VEP}}}_{m^*} = V^*_{m^*} \quad \forall \pi^*_{m_{VEP}} \in \Pi_{VEP}, \tag{8}$$

which implies:

$$\left\| V^*_{m^*} - V^{\pi^*_{m_{VEP}}}_{m^*} \right\|_\infty = 0 \quad \forall \pi^*_{m_{VEP}} \in \Pi_{VEP}. \tag{9}$$

$\square$

**Theorem 2.** *Let $m_{VEP} \in \mathcal{M}_{VEP}$ be a VE partial model of the true environment $m^* \in \mathcal{M}$, and let $\tilde{m}_{VEP} \in \mathcal{M}_{VEP}$ be model that comprises of the reward function of $m_{VEP}$ and a transition distribution that is estimated from $n$ samples for each $(f, a)$ pair. Let $\Pi_{r_{VEP}} \equiv \{\pi \mid \exists\, p_{VEP}\ s.t\ \pi\ is\ optimal\ in\ (p_{VEP}, r_{VEP})\}$. Then, certainty-equivalence planning with $\tilde{m}_{VEP}$ has planning loss:*

$$\left\| V^*_{m_{VEP}} - V^{\pi^*_{\tilde{m}_{VEP}}}_{m_{VEP}} \right\|_\infty \le \frac{2R_{\max}}{(1-\gamma)^2} \sqrt{\frac{1}{2n} \log \frac{2|\mathcal{F}_{VEP}||\mathcal{A}||\Pi_{r_{VEP}}|}{\delta}}, \tag{10}$$

*with probability at least $1 - \delta$.*

*Proof.* Similar to Jiang et al. (2015), we prove Theorem 2 with two lemmas: Lemma 1 translates planning loss to value error, and Lemma 2 relates value error to a Bellman-residual-like quantity that has a uniform deviation bound which depends on $|\Pi_{r_{VEP}}|$.

**Lemma 1.** *For any $\tilde{m}_{VEP} = (\tilde{p}_{VEP}, \tilde{r}_{VEP})$ with $\tilde{r}_{VEP}$ bounded by $[0, R_{\max}]$,*

$$\left\| V^*_{m_{VEP}} - V^{\pi^*_{\tilde{m}_{VEP}}}_{m_{VEP}} \right\|_\infty \le 2 \max_{\pi: \mathcal{F} \to \mathcal{A}} \left\| V^\pi_{m_{VEP}} - V^\pi_{\tilde{m}_{VEP}} \right\|_\infty. \tag{11}$$

*In particular, if $\tilde{r}_{VEP} = r_{VEP}$, we have*

$$\left\| V^*_{m_{VEP}} - V^{\pi^*_{\tilde{m}_{VEP}}}_{m_{VEP}} \right\|_\infty \le 2 \max_{\pi \in \Pi_{r_{VEP}}} \left\| V^\pi_{m_{VEP}} - V^\pi_{\tilde{m}_{VEP}} \right\|_\infty. \tag{12}$$

*Proof.* $\forall f \in \mathcal{F}_{VEP}$,

$$V^{\pi^*_{m_{VEP}}}_{m_{VEP}}(f) - V^{\pi^*_{\tilde{m}_{VEP}}}_{m_{VEP}}(f) = \left( V^{\pi^*_{m_{VEP}}}_{m_{VEP}}(f) - V^{\pi^*_{m_{VEP}}}_{\tilde{m}_{VEP}}(f) \right) - \left( V^{\pi^*_{\tilde{m}_{VEP}}}_{m_{VEP}}(f) - V^{\pi^*_{\tilde{m}_{VEP}}}_{\tilde{m}_{VEP}}(f) \right) \tag{13}$$

$$+ \left( V^{\pi^*_{m_{VEP}}}_{\tilde{m}_{VEP}}(f) - V^{\pi^*_{\tilde{m}_{VEP}}}_{\tilde{m}_{VEP}}(f) \right)$$

$$\le \left( V^{\pi^*_{m_{VEP}}}_{m_{VEP}}(f) - V^{\pi^*_{m_{VEP}}}_{\tilde{m}_{VEP}}(f) \right) - \left( V^{\pi^*_{\tilde{m}_{VEP}}}_{m_{VEP}}(f) - V^{\pi^*_{\tilde{m}_{VEP}}}_{\tilde{m}_{VEP}}(f) \right) \tag{14}$$

$$\le 2 \max_{\pi \in \left\{ \pi^*_{m_{VEP}}, \pi^*_{\tilde{m}_{VEP}} \right\}} |V^\pi_{m_{VEP}}(f) - V^\pi_{\tilde{m}_{VEP}}(f)|. \tag{15}$$

Eqn. 11 follows from taking the max over all feature vectors on both sides of the inequality and noticing that the set of all policies is a trivial superset of $\left\{ \pi^*_{m_{VEP}}, \pi^*_{\tilde{m}_{VEP}} \right\}$. If $\tilde{r}_{VEP} = r_{VEP}$, the bound can be tightened since $\left\{ \pi^*_{m_{VEP}}, \pi^*_{\tilde{m}_{VEP}} \right\} \in \Pi_{r_{VEP}}$, and Eqn. 12 follows. $\square$

**Lemma 2.** *For any $\tilde{m}_{VEP} = (\tilde{p}_{VEP}, \tilde{r}_{VEP})$ with $\tilde{r}_{VEP}$ bounded by $[0, R_{\max}]$, $\forall \pi: \mathcal{F}_{VEP} \to \mathcal{A}$,*

$$\left\| Q^\pi_{m_{VEP}} - Q^\pi_{\tilde{m}_{VEP}} \right\|_\infty \le \frac{1}{1-\gamma} \max_{f \in \mathcal{F}_{VEP}, a \in \mathcal{A}} \left| \tilde{r}_{VEP}(f, a) + \gamma \langle \tilde{p}_{VEP}(f, a, \cdot), V^\pi_{m_{VEP}} \rangle - Q^\pi_{m_{VEP}}(f, a) \right|. \tag{16}$$

*Proof.* Given any policy $\pi$, define action value functions such that $Q_0, Q_1, \ldots, Q_n, \ldots$ such that $Q_0 = Q_{m_{VEP}}^\pi$, and

$$Q_n(f, a) = \tilde{r}_{VEP}(f, a) + \gamma \langle \tilde{p}_{VEP}(f, a, \cdot), V_{n-1} \rangle, \tag{17}$$

where $V_{n-1}(f) = Q_{n-1}(f, \pi(f))$. Notice that

$$\|Q_n - Q_{n-1}\|_\infty = \gamma \max_{f \in \mathcal{F}_{VEP}, a \in \mathcal{A}} |\langle \tilde{p}_{VEP}(f, a, \cdot), (V_{n-1} - V_{n-2}) \rangle| \tag{18}$$

$$\leq \gamma \max_{f \in \mathcal{F}_{VEP}, a \in \mathcal{A}} \|\tilde{p}_{VEP}(f, a, \cdot)\|_1 \|V_{n-1} - V_{n-2}\|_\infty \tag{19}$$

$$= \gamma \|V_{n-1} - V_{n-2}\|_\infty \tag{20}$$

$$\leq \gamma \|Q_{n-1} - Q_{n-2}\|_\infty, \tag{21}$$

so

$$\|Q_n - Q_0\|_\infty \leq \sum_{k=0}^{n-1} \|Q_{k+1} - Q_k\|_\infty \tag{22}$$

$$\leq \|Q_1 - Q_0\|_\infty \sum_{k=0}^{n-1} \gamma^{k-1}. \tag{23}$$

Taking the limit of $n \to \infty$, $Q_n \to Q_{\tilde{m}_{VEP}}^\pi$, and we have,

$$\left\| Q_{\tilde{m}_{VEP}}^\pi - Q_0 \right\|_\infty \leq \frac{1}{1 - \gamma} \|Q_1 - Q_0\|_\infty. \tag{24}$$

This completes the proof, noticing that $Q_0 = Q_{m_{VEP}}^\pi$, $V_0 = V_{m_{VEP}}^\pi$, and $Q_1(f, a) = \tilde{r}_{VEP}(f, a) + \gamma \langle \tilde{p}_{VEP}(f, a, \cdot), V_{m_{VEP}}^\pi \rangle$. $\square$

From Eqn. 12 in Lemma 1 and Lemma 2, we have

$$\left\| V_{m_{VEP}}^* - V_{m_{VEP}}^{\pi_{\tilde{m}_{VEP}}^*} \right\|_\infty \leq 2 \max_{\pi \in \Pi_{r_{VEP}}} \left\| V_{m_{VEP}}^\pi - V_{\tilde{m}_{VEP}}^\pi \right\|_\infty \tag{25}$$

$$\leq 2 \max_{\pi \in \Pi_{r_{VEP}}} \left\| Q_{m_{VEP}}^\pi - Q_{\tilde{m}_{VEP}}^\pi \right\|_\infty \tag{26}$$

$$= 2 \max_{f \in \mathcal{F}_{VEP}, a \in \mathcal{A}, \pi \in \Pi_{r_{VEP}}} \left| Q_{m_{VEP}}^\pi(f, a) - Q_{\tilde{m}_{VEP}}^\pi(f, a) \right|_\infty \tag{27}$$

$$\leq \frac{2}{1 - \gamma} \max_{f \in \mathcal{F}_{VEP}, a \in \mathcal{A}, \pi \in \Pi_{r_{VEP}}} \left| \tilde{r}_{VEP}(f, a) + \gamma \langle \tilde{p}_{VEP}(f, a, \cdot), V_{m_{VEP}}^\pi \rangle - Q_{m_{VEP}}^\pi(f, a) \right|. \tag{28}$$

For any particular $f$, $a$, $\pi$ tuple, according to Hoeffding's inequality, $\forall t > 0$,

$$p \left( \left| \tilde{r}_{VEP}(f, a) + \gamma \langle \tilde{p}_{VEP}(f, a, \cdot), V_{m_{VEP}}^\pi \rangle - Q_{m_{VEP}}^\pi(f, a) \right| > t \right) \leq 2 \exp \left( -\frac{2nt^2}{R_{\max}^2/(1 - \gamma)^2} \right), \tag{29}$$

as $\tilde{r}_{VEP}(f, a) + \gamma \langle \tilde{p}_{VEP}(f, a, \cdot), V_{m_{VEP}}^\pi \rangle$ is the average of i.i.d. samples bounded in $[0, R_{\max}/(1 - \gamma)]$, with mean $Q_{m_{VEP}}^\pi(f, a)$. To obtain a uniform bound over all $(f, a, \pi)$ tuples, we set the right-hand side of Eqn. 29 to $\delta/|\mathcal{F}_{VEP}||\mathcal{A}||\Pi_{r_{VEP}}|$ and solve for $t$, and the theorem follows. $\square$

**Theorem 3.** *Let $m_{VEP} \in \mathcal{M}_{VEP}$ be a VE partial model of the true environment $m^* \in \mathcal{M}$. Let $\tilde{m}_{VEP} \in \mathcal{M}_{VEP}$ be the corresponding approximate VE partial model that has the same reward function as $m_{VEP}$, but whose transition distribution is estimated by $m$ calls to the generative model $m_{VEP}$, where*

$$m = \mathcal{O} \left( \frac{|\mathcal{F}_{VEP}||\mathcal{A}|}{(1 - \gamma)^4 \varepsilon^2} \right), \tag{30}$$

*and let $Q_{\tilde{m}_{VEP}}^k$ be the value returned by Q-value iteration at the kth epoch. Then, with probability greater than $1 - \delta$, the following holds for all $f \in \mathcal{F}_{VEP}$ and $a \in \mathcal{A}$:*

$$\left\| Q_{\tilde{m}_{VEP}}^k - Q_{m_{VEP}}^* \right\|_\infty \leq \varepsilon, \tag{31}$$

*where $k = \frac{\log(\varepsilon(1 - \gamma))}{\log \gamma}$ and $Q_{m_{VEP}}^*$ is the optimal action value function in $m_{VEP}$.*

*Proof.* Before starting the proof, let us first define generative models. A *generative model*, or a *sampler*, is a model that can provide us with samples $f' \sim p(f, a, \cdot)$ for all $f \in \mathcal{F}_{VEP}$ and $a \in \mathcal{A}$. Now that we have defined generative models, let us assume we have access to a generative model $m_{VEP}$ and suppose we call our this model $N$ times at each $(f, a)$ pair. Let $\hat{p}$ be the transition distribution of our empirical model, defined as follows:

$$\hat{p}(f, a, f') = \frac{\text{count}(f, a, f')}{N} = \frac{\sum_{i=1}^{N} \mathbb{I}_{f'_i = f'}}{N}, \tag{32}$$

where $f_i \sim p(f, a, \cdot)$, $\forall i \in \{1, \ldots, N\}$, and $\text{count}(f, a, f')$ is the number of times the pair $(f, a)$ transitions to $f'$.

Moving on the main proof, by adding and subtracting $Q_{\tilde{m}_{VEP}}^{\pi_{\tilde{m}_{VEP}}^*}$, we can rewrite $Q_{\tilde{m}_{VEP}}^k - Q_{m_{VEP}}^*$ as follows:

$$Q_{\tilde{m}_{VEP}}^k - Q_{m_{VEP}}^* = \underbrace{Q_{\tilde{m}_{VEP}}^k - Q_{\tilde{m}_{VEP}}^{\pi_{\tilde{m}_{VEP}}^*}}_{(i)} + \underbrace{Q_{\tilde{m}_{VEP}}^{\pi_{\tilde{m}_{VEP}}^*} - Q_{m_{VEP}}^*}_{(ii)} \tag{33}$$

Bounding Term $(i)$:

$$\left\| Q_{\tilde{m}_{VEP}}^k - Q_{\tilde{m}_{VEP}}^{\pi_{\tilde{m}_{VEP}}^*} \right\|_\infty = \max_{f \in \mathcal{F}_{VEP}, a \in \mathcal{A}} \left| r_{VEP}(f, a) + \gamma \tilde{p}_{VEP} V_{\tilde{m}_{VEP}}^{k-1}(f, a) - \left( r_{VEP}(f, a) + \gamma \tilde{p}_{VEP} V_{\tilde{m}_{VEP}}^{\pi_{\tilde{m}_{VEP}}^*}(f, a) \right) \right| \tag{34}$$

$$= \max_{f \in \mathcal{F}_{VEP}, a \in \mathcal{A}} \gamma \left| \tilde{p}_{VEP} \left( V_{\tilde{m}_{VEP}}^{k-1} - V_{\tilde{m}_{VEP}}^{\pi_{\tilde{m}_{VEP}}^*} \right)(f, a) \right| \tag{35}$$

$$\leq \gamma \left\| V_{\tilde{m}_{VEP}}^{k-1} - V_{\tilde{m}_{VEP}}^{\pi_{\tilde{m}_{VEP}}^*} \right\|_\infty \tag{36}$$

$$\leq \gamma \max_{f \in \mathcal{F}_{VEP}} \left| \max_{a \in \mathcal{A}} Q_{\tilde{m}_{VEP}}^{k-1}(f, a) - \max_{a \in \mathcal{A}} Q_{\tilde{m}_{VEP}}^{\pi_{\tilde{m}_{VEP}}^*}(f, a) \right| \tag{37}$$

$$\leq \gamma \max_{f \in \mathcal{F}_{VEP}, a \in \mathcal{A}} \left| Q_{\tilde{m}_{VEP}}^{k-1}(f, a) - Q_{\tilde{m}_{VEP}}^{\pi_{\tilde{m}_{VEP}}^*}(f, a) \right| \tag{38}$$

$$= \gamma \left\| Q_{\tilde{m}_{VEP}}^{k-1} - Q_{\tilde{m}_{VEP}}^{\pi_{\tilde{m}_{VEP}}^*} \right\|_\infty. \tag{39}$$

Unrolling the last inequality $k$ times, we obtain:

$$\left\| Q_{\tilde{m}_{VEP}}^k - Q_{\tilde{m}_{VEP}}^{\pi_{\tilde{m}_{VEP}}^*} \right\|_\infty \leq \gamma^k \| Q_{\tilde{m}_{VEP}}^0 - Q_{\tilde{m}_{VEP}}^{\pi_{\tilde{m}_{VEP}}^*} \| \tag{40}$$

$$\leq \frac{\gamma^k}{1 - \gamma}. \tag{41}$$

Bounding Term $(ii)$:

$$\left( Q_{\tilde{m}_{VEP}}^{\pi_{\tilde{m}_{VEP}}^*} - Q_{m_{VEP}}^* \right)(f, a) = \gamma \tilde{p}_{VEP} V_{\tilde{m}_{VEP}}^{\pi_{\tilde{m}_{VEP}}^*}(f, a) - \gamma p_{VEP} V_{m_{VEP}}^*(f, a) \tag{42}$$

$$= \gamma \left( \tilde{p}_{VEP} - p_{VEP} \right) V_{m_{VEP}}^*(f, a) - \gamma \tilde{p}_{VEP} \left( V_{\tilde{m}_{VEP}}^{\pi_{\tilde{m}_{VEP}}^*} - V_{m_{VEP}}^* \right)(f, a) \tag{43}$$

$$= \gamma \left( \tilde{p}_{VEP} - p_{VEP} \right) V_{m_{VEP}}^*(f, a) \tag{44}$$

$$- \gamma \sum_{f' \in \mathcal{F}} \tilde{p}_{VEP}(f, a, f') (\max_{a' \in \mathcal{A}} Q_{\tilde{m}_{VEP}}^{\pi_{\tilde{m}_{VEP}}^*}(f', a') - \max_{a' \in \mathcal{A}} Q_{m_{VEP}}^*(f', a')).$$

Therefore,

$$\left\| Q_{\tilde{m}_{VEP}}^{\pi_{\tilde{m}_{VEP}}^*} - Q_{m_{VEP}}^* \right\|_\infty \leq \gamma \max_{f \in \mathcal{F}_{VEP}, a \in \mathcal{A}} \left| (\tilde{p}_{VEP} - p_{VEP}) V_{m_{VEP}}^*(f, a) \right| + \gamma \left\| Q_{\tilde{m}_{VEP}}^{\pi_{\tilde{m}_{VEP}}^*} - Q_{m_{VEP}}^* \right\|_\infty \tag{45}$$

$$\leq \frac{\gamma}{1 - \gamma} \left\| (\tilde{p}_{VEP} - p_{VEP}) V_{m_{VEP}}^* \right\|_\infty. \tag{46}$$

Fix a $(f, a)$ pair:

$$(\tilde{p}_{VEP} - p_{VEP}) V^*_{m_{VEP}} = \frac{1}{N} \sum_{i=1}^{N} V^*_{m_{VEP}}(f'_i) - E_{f' \in p_{VEP}(f,a,f')} \left[ V^*_{m_{VEP}}(f') \right] \tag{47}$$

$$= \frac{1}{N}(S_N - E[S_N]), \tag{48}$$

where $S_N = \sum_{i=1}^{N} X_i$ and $X_i = V^*_{m_{VEP}}(f'_i)$. $X_i$ are random independent variables and $|X_i| \leq \frac{1}{1-\gamma}$. Applying Hoeffding's inequality, we obtain $\forall t > 0$:

$$p\left( \frac{1}{N}(S_N - E[S_N]) \geq t \right) \leq 2 \exp\left( \frac{-N^2 t^2}{N/(1-\gamma)^2} \right) \tag{49}$$

$$= 2 \exp\left( -N t^2 (1-\gamma)^2 \right) \tag{50}$$

$$p\left( \max_{f \in \mathcal{F}_{VEP}, a \in \mathcal{A}} \left| (\tilde{p}_{VEP} - p_{VEP}) V^*_{m_{VEP}}(f, a) \right| \geq t \right) = p\left( \exists (f, a) \text{ s.t. } \left| (\tilde{p}_{VEP} - p_{VEP}) V^*_{m_{VEP}}(f, a) \right| \geq t \right) \tag{51}$$

$$\leq \sum_{f \in \mathcal{F}, a \in \mathcal{A}} p\left( \left| (\tilde{p}_{VEP} - p_{VEP}) V^*_{m_{VEP}}(f, a) \right| \geq t \right)$$

(Union Bound)

$$= 2 |\mathcal{F}_{VEP}||\mathcal{A}| \exp\left( -N t^2 (1-\gamma)^2 \right) \tag{52}$$

Let the failure probability $\delta > 0$. Solve for $t$,

$$2 |\mathcal{F}_{VEP}||\mathcal{A}| \exp\left( -N t^2 (1-\gamma)^2 \right) = t \tag{53}$$

$$\Rightarrow t = \frac{1}{1-\gamma} \sqrt{\frac{\log(2|\mathcal{F}_{VEP}||\mathcal{A}|/\delta)}{N}}. \tag{54}$$

With probability at least $1 - \delta$,

$$\left\| Q^{\pi^*_{\tilde{m}_{VEP}}}_{\tilde{m}_{VEP}} - Q^*_{m_{VEP}} \right\|_\infty \leq \frac{\gamma}{1-\gamma} \max_{f \in \mathcal{F}_{VEP}, a \in \mathcal{A}} \left\| (\tilde{p}_{VEP} - p_{VEP}) V^*_{m_{VEP}} \right\|_\infty \tag{55}$$

$$\leq \frac{\gamma}{(1-\gamma)^2} \sqrt{\frac{\log(2|\mathcal{F}_{VEP}||\mathcal{A}|/\delta)}{N}}. \tag{56}$$

We conclude

$$\left\| Q^k_{\tilde{m}_{VEP}} - Q^*_{m_{VEP}} \right\|_\infty \leq \left\| Q^k_{\tilde{m}_{VEP}} - Q^{\pi^*_{\tilde{m}_{VEP}}}_{\tilde{m}_{VEP}} \right\|_\infty + \left\| Q^{\pi^*_{\tilde{m}_{VEP}}}_{\tilde{m}_{VEP}} - Q^*_{m_{VEP}} \right\|_\infty \tag{57}$$

$$\leq \frac{\gamma^k}{(1-\gamma)} + \frac{\gamma}{(1-\gamma)^2} \sqrt{\frac{\log(2|\mathcal{F}_{VEP}||\mathcal{A}|/\delta)}{N}}. \tag{58}$$

By choosing

$$k = \frac{\log(2(1-\gamma)/\varepsilon)}{\log \gamma}$$

and

$$N = \frac{4\gamma^2}{(1-\gamma)^4 \varepsilon^2} \log(2|\mathcal{F}_{VEP}||\mathcal{A}|/\delta),$$

we get $\left\| Q^k_{\tilde{m}_{VEP}} - Q^*_{m_{VEP}} \right\|_\infty \leq \varepsilon/2 + \varepsilon/2 = \varepsilon$. Therefore, the total number of samples (calls to the generative model) to get an $\varepsilon$ estimation of the optimal $Q$-value is:

$$N|\mathcal{F}_{VEP}||\mathcal{A}| = \mathcal{O}\left( \frac{|\mathcal{F}_{VEP}||\mathcal{A}|}{(1-\gamma)^4 \varepsilon^2} \right). \tag{59}$$

$\square$

## B   ALGORITHM PSEUDOCODES

---

**Algorithm 1** Model-Based Q-Value Iteration

---

1: Initialize the parameters $V^0 = 0$ and $Q^0 = 0$
2: **for** episode $k = 1, \ldots, K$ **do**
3:     **for** $(f, a) \in \mathcal{F} \times \mathcal{A}$ **do**
4:         $Q^k(f, a) = r(f, a) + \gamma \tilde{p} V^{k-1}(f, a)$
5:         $V^k(f) = \max_{a \in \mathcal{A}} Q^k(f, a)$
6:     **end for**
7: **end for**
8: **Return** $Q^K$

---

---

**Algorithm 2** The Straight-Forward Decision-Time Planning Algorithm of Zhao et al. (2021)

---

1: Initialize the parameters $\theta, \eta$ & $\omega$ of $\phi_\theta : \mathcal{S} \to \mathcal{F}, Q_\eta : \mathcal{F} \times \mathcal{A} \to \mathbb{R}$ & $m_\omega = (p_\omega, r_\omega)$
2: Initialize the replay buffer $\mathcal{B} \leftarrow \{\}$
3: $N_{ple} \leftarrow$ number of episodes to perform planning and learning
4: $N_{rbt} \leftarrow$ number of samples that the replay buffer must hold to perform planning and learning
5: $n_s \leftarrow$ number of time steps to perform search
6: $n_{bs} \leftarrow$ number of samples to sample from the replay buffer
7: $h \leftarrow$ search heuristic
8: $T \leftarrow$ replay buffer sampling strategy
9: $i \leftarrow 0$
10: **while** $i < N_{ple}$ **do**
11:     $S \leftarrow$ reset environment
12:     **while** not done **do**
13:         $A \leftarrow \epsilon$-greedy(tree_search_with_bootstrapping($\phi_\theta(S), m_\omega, Q_\eta, n_s, h$))
14:         $R, S',$ done $\leftarrow$ environment($A$)
15:         $\mathcal{B} \leftarrow \mathcal{B} + \{(S, A, R, S', \text{done})\}$
16:         **if** $|\mathcal{B}| \geq N_{rbt}$ **then**
17:             $\mathcal{D} \leftarrow$ sample_batch($\mathcal{B}, n_{bs}, T$)
18:             Update $\phi_\theta, Q_\eta$ & $m_\omega$ with $\mathcal{D}$
19:         **end if**
20:         $S \leftarrow S'$
21:     **end while**
22:     $i \leftarrow i + 1$
23: **end while**
24: **Return** $\phi_\theta, Q_\eta$ & $m_\omega$

---

Note that Alg. 2[2] does not employ the "bottleneck mechanism" introduced in (Zhao et al., 2021).

## C   EXPERIMENTAL DETAILS

In this section, we provide the implementation details of the environments that are used in Sec. 5 together with the details of the models that are used in the scalability experiments of Sec. 5.1. We also provide the implementation details of the straightforward decision-time planning algorithm of Zhao et al. (2021) that was used in Sec. 5.2.

### C.1   IMPLEMENTATION DETAILS OF THE SW ENVIRONMENT

As stated in Sec. 3, in the Squirrel's World (SW) environment the squirrel's job is to navigate from cell E1 (its initial state) to cell E16 (the terminal state) to pickup the nut without getting caught by

---

[2]See `https://github.com/mila-iqia/Conscious-Planning` for the publicly available actual code.

the hawk that flies back and forth horizontally along row C. At each time step, the squirrel receives as input an $5 \times 16$ image of the current state of the environment and then, through the use of a pre-defined state encoder, transforms this image into a feature vector that contains information regarding all aspects of the current state of the environment, i.e., the feature vector contains information on the current position of the squirrel and the cloud, the current wind direction in rows A and B, the current position and direction of the hawk and the current weather condition. Based on this, the squirrel selects an action that either moves it to the left or right cell, or keeps it position fixed (except if the agent is trying to move out of the boundaries of the world in which case its position is kept constant). If the squirrel gets caught by the hawk or if it is out of time, it receives a reward of 0 and the episode terminates, and if the squirrel successfully navigates to the nut within the given time limit, it gets a reward of $+10$ and the episode terminates. The agent-environment interaction lasts for 100 time steps, after which the agent receives a done signal, marking the end of the episode.

## C.2   IMPLEMENTATION DETAILS OF THE 2RDO ENVIRONMENTS

In the 2RDO environments, the agent, depicted by the red triangle, spawns in top-left of the top room and has to navigate to the green goal cell located in the bottom-right of the same room, regardless of the gaseous motions of the obstacles in the bottom room. Here, at each time step, the obstacles move to one of its neighboring cells (except if it is trying to move out of the boundaries of the world in which case its position is kept constant). At each time step, the agent receives an image of the current state of the grid and then, through the use of a learned state encoder, transforms this image into a feature vector. Based on this, the agent selects an action that either turns it left or right, or moves it forward (except if the agent is trying to move out of the boundaries of the world in which case its position is kept constant). If the agent successfully navigates to the goal cell within the given time limit, it receives a reward of $+1$ and the episode terminates. The agent-environment interaction lasts for 50 time steps for the 8x8 environments and 100 time steps for the 16x16 environments, after which the agent receives a done signal, marking the end of the episode.

## C.3   DETAILS OF THE HAND-ENGINEERED MODELS

The details of what the models in Sec. 5.1 model can be found in Table 1.

Table 1: Several non-VE and VE partial models of the SW environment.

| | |
|---|---|
| $m_1$ | squirrel position, cloud position |
| $m_2$ | squirrel position, cloud position, wind direction |
| $m_3$ | squirrel position, cloud position, wind direction, hawk position |
| $m_4$ | squirrel position, hawk position, hawk direction |
| $m_5$ | squirrel position, hawk position, hawk direction, cloud position |
| $m_6$ | squirrel position, hawk position, hawk direction, cloud position, wind direction |
| $m_7$ | squirrel position, hawk position, hawk direction, cloud position, wind direction, weather |

## C.4   DETAILS AND HYPERPARAMETERS OF THE DECISION-TIME PLANNING ALGORITHM

The details and hyperparameters of the straightforward decision-time of Zhao et al. (2021) that we have used can be found in Table 2.

Table 2: Details and hyperparameters of Alg. 2.

| | |
|---|---|
| $\phi_\theta$ | A regular neural network feature extractor |
| $Q_\eta$ | A regular neural network |
| $m_\omega$ | A regular neural network |
| $N_{ple}$ | 50M |
| $N_{rbt}$ | 50k |
| $n_s$ | 20 |
| $n_{bs}$ | 128 |
| $h$ | best-first search (training), random search (evaluation) |
| $T$ | random sampling |
| $\epsilon$ | linearly decays from 1.0 to 0.0 over the first 1M time steps |

For more details (such as the NN architectures, replay buffer sizes, learning rates, exact details of the tree search, . . . ), we refer the reader to the publicly available code and the supplementary material of Zhao et al. (2021).

## C.5 DETAILS OF THE ENCODER SHAPING PROCEDURE DURING TRAINING

In Sec. 5.2, we argued that one of the important inductive biases that is likely to guide the agent in coming up with only the relevant features of the environment is to only let the value estimator shape the encoder and to prevent the model from doing so. This is pictorially depicted in Fig. 7.

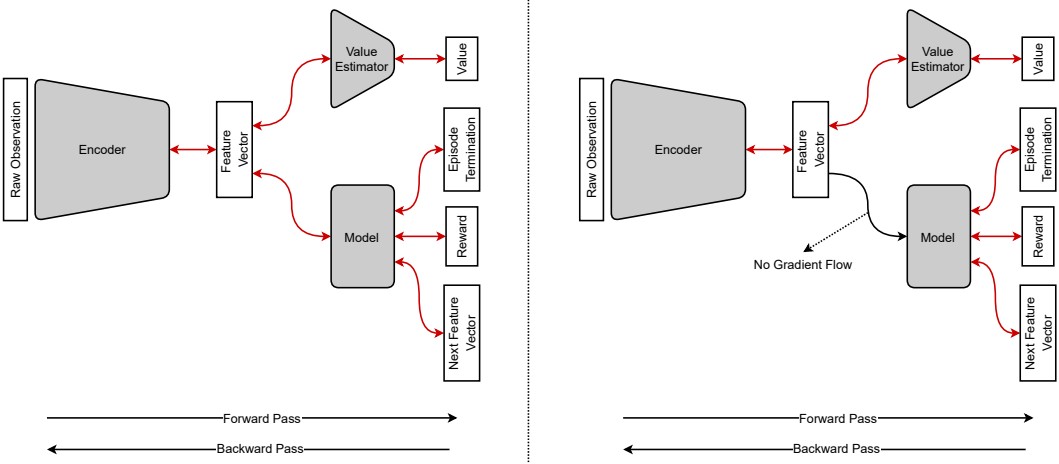

Figure 7: A pictorial representation of how the agent can be trained so that it can come up with relevant features of the environment. (Right) The regular way of training, (Left) the way it can be done.

