# OpenReview forum: "Minimal Value-Equivalent Partial Models for Scalable and Robust Planning in Lifelong Reinforcement Learning"
_ICLR.cc/2023/Conference — Submitted to ICLR 2023_

### Official Review · Reviewer_oVRM · 2022-10-19

**Confidence:** 4
**Correctness:** 4
**Technical Novelty And Significance:** 2
**Empirical Novelty And Significance:** 2
**Recommendation:** 3

**Clarity, Quality, Novelty And Reproducibility:**

The writing of the paper is clear, and the experiments seem reproducible.
The novelty of the paper is limited as the positive effect of removing irrelevant features is well-known and studied in general machine learning. However, the paper does neither provide a new approach to a solution nor provides a very deep theoretical analysis.


**Strength And Weaknesses:**

Strong points:
* The paper defines all concepts step by step and the results are clear

Weak points:
* The paper's contribution is limited as it formalizes a relatively well-known effect.
* The paper argues much with MDPs where function approximation might not even be necessary.



**Summary Of The Paper:**

The paper examines the advantages of using minimal value equivalent feature sets for state representation compared to feature spaces including irrelevant features. Value-equivalence is given if a reduced feature set leads to the same equivalent policies. The paper performs some experiments comparing partial models to complete models on basic environments with limited state and action spaces.


**Summary Of The Review:**

All in all, the contribution of the paper is limited as it basically examines the effects of optimal subsets which are obvious if they can be determined. However, the paper does not really help to answer questions on how to find representation spaces which are value-equivalent. In addition, there is a strong link to causal analysis in MDPs. Causality analysis basically yields a framework for analysing whether features yield a dependency in estimating a distribution.

---

> ### Author Response · Authors · 2022-11-07
> **Response to Reviewer oVRM (1)**
>
> We would like to thank the reviewer for their feedback on the paper.
>
> **Responses to the Weaknesses part of the reviewer’s review:**
>
> 1. “The paper’s contribution is limited as it formalizes a relatively well-known effect”
>     - We are not sure how to respond to this concern as it is **not clear** what a “relatively well-known affect” is. Can the reviewer elaborate more on this?
>     - From reading other parts of the review, we assume that the “relatively well-known effect” is the “positive effect of removing irrelevant features”. If this is the case, we do **not** agree that this is a relatively well-known effect in reinforcement learning in general and model-based reinforcement learning in particular. If the reviewer is aware of particular studies that do this, we would be glad to know about them.
>
> 2. “The paper argues much with MDPs where function approximation might not even be necessary.”
>     - We are not sure what this concern means. We would appreciate it if the reviewer can elaborate more on it. As far as we are aware, reinforcement learning tasks are formalized through MDPs **regardless** of whether function approximation is used or not by the agent.
>     - If the main concern of the reviewer is that the proposed type of models are not well-suited to use with function approximation, we would like to note that the models that we define through definitions 1 to 3 are all over the **feature space** that is generated by the agent’s encoder (which is obtained through the use of function approximation). So, minimal VE models are **definitely well-suited to use with function approximation**. We kindly refer the reviewer to the “Reinforcement Learning” section of Sec. 2 to learn more about the specific framework we are considering in this paper.
>
>
> **Responses to the Clarity, Quality, Novelty and Reproducibility part of the reviewer’s review:**
>
> 1. “The novelty of the paper is limited as the positive effect of removing irrelevant features is well-known and studied in general machine learning. However, the paper does neither provide a new approach to a solution nor provides a very deep theoretical analysis.”
>     - Regarding the novelty, as stated above, we do **not** think that the “positive effect of removing irrelevant features” is a well-known and studied concept in the reinforcement learning literature in general and the model-based reinforcement learning literature in particular. If the reviewer does not agree with us, we would be interested in pointers to studies that study this concept. We would be happy to incorporate them in our related work section.
>     - “However, the paper does neither provide a new approach to a solution” We are not sure if this is the case. In this paper, we introduce minimal VE partial models as a **solution** to perform scalable and robust planning in lifelong reinforcement learning scenarios.
>     - Would the reviewer be able to clarify what he/she means by “very deep theoretical analysis”? As far as we are aware, our study does provide a **fair amount of theory** in Sec. 4, providing value and planning loss analyses and computational and sample complexity analyses.
>
>
> **Responses to the Summary part of the reviewer’s review:**
>
> 1. “However, the paper does not really help to answer questions on how to find representation spaces which are value-equivalent.”
>     - Actually the paper is **not** concerned on how to find representation spaces that are **value-equivalent**. There is a whole recent literature that tries to deal with this question (we refer to the Value-Equivalence section of Sec. 6 for more details).
>     - In this paper, we are interested in finding representations for **minimal** VE partial models and we **provide** two simple heuristics in doing so (see Question 6) and we have also **demonstrated** their use through careful experiments. We note in the conclusion that these heuristics are not principled methods, however, we note that it is mainly a problem in the representation learning literature and not a problem with minimal VE partial models itself.
>
> 2. “In addition, there is a strong link to causal analysis in MDPs. Causality analysis basically yields a framework for analysing whether features yield a dependency in estimating a distribution.”
>    - We are not sure if this is the case (at least in the **current** form of this study) as minimal VE partial models only model the relevant aspects of the agent’s environment **without explicitly** dealing with the causal relationships between the feature’s in the models. That being said, it is definitely possible to extend this work toward the direction of learning minimal VE partial models that explicitly focus on the causal relationships between features.
>     - We would like to thank the reviewer for providing this **constructive** feedback as it is definitely an interesting direction to look up in the future.

---

> > ### Author Response · Authors · 2022-11-07
> > **Response to Reviewer oVRM (2)**
> >
> > **Responses to the Correctness part of the reviewer’s review:**
> >
> > In the correctness part of the review it states that “1: The main claims of the paper are incorrect or not at all supported by theory or empirical results.”. We would like to note that we do **not** agree with this. Neither do we make claims that are incorrect in this paper nor do we leave them unsupported by theory or experiments. Could the reviewer explain the **reasoning** behind this statement?

---

> > > ### Comment · Reviewer_oVRM · 2022-11-11
> > > **correctness**
> > >
> > > I have to apologize here and I already corrected the review w.r.t this point. Besides my other concerns, I agree that I did not find any incorrectness in your claims. It seems that I did not look well enough and confused the order of the ranking. In other reviewing systems I work with in parallel, the options start in the opposite order.
> > > As mentioned above I still have some concerns about the experiments though.

---

> > > > ### Author Response · Authors · 2022-11-12
> > > > **Response to correctness**
> > > >
> > > > We would like to thank the reviewer for correcting this point. Things like this can happen all the time, no problem.

---

> > ### Comment · Reviewer_oVRM · 2022-11-11
> > **comment on author response**
> >
> > Dear authors,
> >
> > thank you for your comments on the review.
> >
> > I apologize that my review was not more precise w.r.t. my concerns about your work. The reviewing period was comparably short this time, and thus, I did not have a lot of time to explain my concerns more deeply.
> >
> > Explanation on the comments:
> > 1. “The paper’s contribution is limited as it formalizes a relatively well-known effect.”
> > Your setting is equivalent to factored MDPs,i.e., where the state is described by a set of factor variables or features. There exist algorithms, e.g. SPUDD (https://arxiv.org/abs/1301.6704), which exploit the summarization of states having the same set of relevant variables. As these "value equivalent" states are jointly optimized, convergence is faster. Thus, the effect is well-understood on the MDP level, and the paper should have put more attention on the question of what happens if the MDP is unknown but learned during training. Furthermore, the relation to the life-long setting is also not highlighted enough. In particular, the question is whether features being in the minimal value equivalent setting in one environment are still to be excluded after the environment adapts. Why can it be ruled out that environments change in a way that rewards or transitions now require these features.
> >
> > 2. “The paper argues much with MDPs where function approximation might not even be necessary.”
> > This refers to the first part of the evaluation. It is not surprising that convergence is still stable if you remove irrelevant features. But the aspect of applying these ideas to learned models and the impact of imperfect models and changing feature spaces during training should be discussed more deeply. The effects of solving the MDP are known from factored MDPs.
> >
> > 3. "However, the paper does not really help to answer questions on how to find representation spaces which are value-equivalent.”
> > In general, you analyze that removing irrelevant features is beneficial but not how to determine which ones should be removed. Admittedly, I overlooked the life-long settings for which you proposed two heuristics. But if these heuristics yield your technical contribution towards finding minimal value equivalent models, it should be introduced in the core sections and not the experiments.
> > If your paper is about finding  *minimal* VE partial models, can you please explain again why the proposed heuristics guarantee minimality? As far as I understood the paper, the heuristics guide the network to ignore irrelevant features but there is no guarantee that the  learned model is minimal.
> >
> >
> > 4. “In addition, there is a strong link to causal analysis in MDPs. Causality analysis basically yields a framework for analysing whether features yield a dependency in estimating a distribution.”
> > The causal relationship which I referred to is rather in the dependency between state features and values (expectation of future rewards) or state features before and after the action.  The later causl relationship is often directly linked to the first one as the agent usually receives rewards when reaching states with particular properties.

---

> > > ### Author Response · Authors · 2022-11-12
> > > **Response to Reviewer oVRM's response**
> > >
> > > We would like to start by thanking the reviewer for taking the time and elaborating more on his/her comments.
> > >
> > > 1. "Your setting is equivalent to factored MDPs,i.e., where the state is described by a set of factor variables or features. There exist algorithms, e.g. SPUDD (https://arxiv.org/abs/1301.6704), which exploit the summarization of states having the same set of relevant variables. As these "value equivalent" states are jointly optimized, convergence is faster."
> > >
> > >     - We are aware that our setting is similar to the factored MDP setting and in fact we have **cited** this framework in the Reinforcement Learning section of the Background section of the paper. Differently, in this study, we are thinking about the **lifelong RL** setting where the environment is large and distribution shifts can happen overtime.
> > >
> > > 2. "Thus, the effect is well-understood on the MDP level, and the paper should have put more attention on the question of what happens if the MDP is unknown but learned during training."
> > >
> > >     - We note that, in this study, **we are actually considering the setting in which MDP (or model) is unknown and it has to be learned from data** (afterall this is a study on model-based RL and not pure planning). In fact in all of our experiments (except for the ones in Q1 and Q3), the MDP (or model) is learned from data (interaction with the environment). We assume that the model is known in Q1 and Q3 as in these questions, we are interested in answer what would be the value loss and computation benefits of minimal VE models if we assume that the agent has access to a perfect model (or MDP) of the environment.
> > >
> > > 3. "Furthermore, the relation to the life-long setting is also not highlighted enough. In particular, the question is whether features being in the minimal value equivalent setting in one environment are still to be excluded after the environment adapts. Why can it be ruled out that environments change in a way that rewards or transitions now require these features."
> > >
> > >     - We would like to thank the reviewer for bringing this up. We tried to emphasize many times in the introduction that this study is about lifelong RL. However, we agree that there this part can be improved further.
> > >     - So, in this study, we are focused lifelong RL scenarios in which the environment is large and mostly contains irrelevant features [1, 2] (much like real-life scenarios in which there are a few relevant aspects, e.g. a fish only has to model what is going on in its local area (like the behavior of its prey and other bigger fish) and does not have model all the complex fluid dynamics or the motion of the other creatures living in land).
> > >     - Another example from the Squirrel’s World environment can be that most of the tasks that are of interest to the Squirrel are always related to the position of the squirrel and the position and direction of the hawk (the squirrel would never have to model e.g. the wind direction in the upper atmosphere). If the squirrel wants to navigate to the middle bushes to obtain another food or to bring the nut back to its starting position, the features that it has learned will be enough to perform these tasks.
> > >     - We realize that our initial submission was not particularly clear on this and thus we have added this detail to the Introduction section of the revised version of the study (see the first paragraph). We are also planning to add more explanation on this to the "illustrative example" part of of Sec. 3 after the rebuttal period.
> > >
> > > 4. "“The paper argues much with MDPs where function approximation might not even be necessary.” This refers to the first part of the evaluation. It is not surprising that convergence is still stable if you remove irrelevant features. But the aspect of applying these ideas to learned models and the impact of imperfect models and changing feature spaces during training should be discussed more deeply. The effects of solving the MDP are known from factored MDPs."
> > >
> > >     - We would like to note that our **theoretical analysis and experiments indeed contains cases in which the model has to be learned from data (so the model is learned and imperfect)** (see the planning loss analysis and sample complexity analysis for the theoretical analysis and Q2, Q4, Q5, Q6, Q7 for the experiments). So we are not sure what the reviewer is referring to. Can the reviewer be more specific?
> > >     - Regarding the changing feature spaces, we would to note that, as stated above, in this study, we are interested in **lifelong RL scenarios where most of the things in the environment are irrelevant and it does not matter if they change or not** (e.g., the squirrel never cares about what kind of physics is involved in the upper atmosphere).

---

> > > > ### Author Response · Authors · 2022-11-12
> > > > **Response to Reviewer oVRM's response (2)**
> > > >
> > > > 5. "However, the paper does not really help to answer questions on how to find representation spaces which are value-equivalent.” In general, you analyze that removing irrelevant features is beneficial but not how to determine which ones should be removed. Admittedly, I overlooked the life-long settings for which you proposed two heuristics. But if these heuristics yield your technical contribution towards finding minimal value equivalent models, it should be introduced in the core sections and not the experiments.
> > > >
> > > >     - We have tried to explain this part as a contribution of the paper in the last paragraph of the Introduction section, however, we agree that this part can be made more clear. We **will clarify it** in the core sections of the study after the rebuttal period.
> > > >
> > > > 6. "If your paper is about finding minimal VE partial models, can you please explain again why the proposed heuristics guarantee minimality? As far as I understood the paper, the heuristics guide the network to ignore irrelevant features but there is no guarantee that the learned model is minimal."
> > > >
> > > >     - We have tried to explain this part in **Q5** as follows: "Even though finding the right inductive biases to train a model-free or model-based RL agent is still an open problem in the representation learning literature (Bengio et al., 2013), in this study, we propose two inductive biases that are likely to guide the agent in coming up with only the relevant features. The first one is to only let the value estimator shape the encoder and prevent the model from doing so (see Fig. 7). In this way, the agent can be guided in learning the features that are relevant for predicting the right values in the environment. And, the second one is to train the agent across a variety of environments in which the irrelevant aspects keep changing and the relevant ones stay the same. In this way, the agent can be guided in not learning the irrelevant aspects of the environment."
> > > >     - Regarding the guarantee on learning only the relevant features, **yes there is indeed no guarantee** as in deep RL we have no control of the representations that are learned by the agent (if we had control it, it wouldn't be deep learning afterall).
> > > >     - But we would like to note that this is **not** a limitation of minimal VE partial models, **but a limitation of the current state of representation learning** as explained in our Conclusion section: "One limitation of our work is that, rather than providing a principled method, we have only provided several heuristics for training deep RL agents that can come up with only the relevant features of the environment. However, we note that this is mainly due to the lack of principled approaches in the representation learning literature, and we believe that this limitation can be overcomed with more principled approaches being introduced in the literature."
> > > >
> > > > 6. "“In addition, there is a strong link to causal analysis in MDPs. Causality analysis basically yields a framework for analysing whether features yield a dependency in estimating a distribution.” The causal relationship which I referred to is rather in the dependency between state features and values (expectation of future rewards) or state features before and after the action. The later causl relationship is often directly linked to the first one as the agent usually receives rewards when reaching states with particular properties."
> > > >
> > > >     - We would like to thank the reviewer for elaborating more on this part. Yes, there is indeed a causal relationship between the state features and values. However, **we are not sure on how to address this concern in the paper**. Could the reviewer be more explicit on what we actions we can take to address it?

---

> > > ### Author Response · Authors · 2022-11-18
> > > **Any outstanding concerns?**
> > >
> > > Dear reviewer, with our response below, we have tried to address your concerns regarding our paper. As there is only one day left for the discussion period to end, we would like to know if you have any outstanding concerns in light of our response? We will be happy to address them.

---

### Official Review · Reviewer_SiWS · 2022-10-27

**Confidence:** 4
**Correctness:** 3
**Technical Novelty And Significance:** 3
**Empirical Novelty And Significance:** 3
**Recommendation:** 3

**Clarity, Quality, Novelty And Reproducibility:**

The paper is well-written and easy to understand. The proposed model seems novel to the best of my knowledge. I have not verified the theorem proofs in the supplementary material, but the empirical results seem to be reproducible.

**Strength And Weaknesses:**

Strengths:

The paper is trying to address a very important research problem and the presented approach is simple to implement which I appreciate a lot. The authors have also presented theoretical results on differences in value-equivalence when learning approximate VE models.

Improvements/Clarifications:

1. What is \delta in Theorem 3? How is it related to \epsilon here?
2. In fig 3, why is 3a only over a single run? Also there are no error bars/variances in 3b and 3c.
3. In section 5.1, the text only talks about models m1-m6. But fig 3 also has m7. What is m7? For fig 3, please show results for all the 7 models in 3a, 3b and 3c? Why only sample 4 out of 7?
4. The results are only for discrete grid-world environments, which makes it unclear if the proposed approach will generalize to more realistic settings.
5. While answering question 7 (on page 9), what does the search budget refer to? Planner lookahead steps or planner time cutoff or something else?
6. The word lifelong is very misleading because it seems to indicate a continual learning flavor, which is not the case. Please consider removing the word from the title and the paper.

**Summary Of The Paper:**

The authors present a framework to define partial models, value-equivalence and a simple method to learn partial models which are minimal and value-equivalent to the underlying MDP. The paper presents empirical results on two grid-world environments.

**Summary Of The Review:**

I can currently only advocate a weak accept given that the experimental evaluation is not strong enough to warrant a better score. I highly recommend removing the word "lifelong" from the paper title, abstract etc. since it is very misleading and feels like the paper is about continual learning.

Edits post-discussion phase:
---------------------------------------
I thank the authors for their clarifications. After more discussion with other reviewers and the AC, I have decided to lower my score to 3: reject, not good enough. I initially kept my original score of 6 following the authors' responses, since I do see value in the work when it comes to learning good partial models of the world. However, the authors have not addressed my concern that the work is not useful for lifelong learning. It currently only deals with a very specific kind of distribution shift which does not require acquiring any new features over time, and hence the model would not truly scale to a lifelong setting. I believe that more thorough experimentation with complex distribution shifts is required to prove that the model would indeed work in the lifelong setting before the work can be accepted.

---

> ### Author Response · Authors · 2022-11-07
> **Response to Reviewer SiWS**
>
> We would like to thank the reviewer for their detailed and partly constructive feedback on the paper. Finding reviews of this kind is rare nowadays, so we really appreciate it.
>
> **Responses to the Weaknesses part of the reviewer’s review:**
>
> 1. “What is \delta in Theorem 3? How is it related to \epsilon here?”
>     - $\delta$ is the failure probability of the inequality in Eqn. 7 and it can be related to $\epsilon$ through the equation that gives $N$ in the proof of Theorem 3. See the equation right before Eqn. 60 in page 15.
>     - However we note that the relation between the two is not important for what we want to show in Theorem 3: the sample complexity benefit of planning with approximate partial VE models.
>
> 2. “In fig 3, why is 3a only over a single run? Also there are no error bars/variances in 3b and 3c.”
>     - Good question. Actually all of the plots in Fig. 3a are over multiple runs. However, as we give enough computational budget to value iteration, it always converges to the same value resulting in plots with no variation.
>     - Regarding Fig. 3b and 3c, we actually had multiple runs, however as the results were too close to each other with close to zero variation, we did not plot them.
>
> 3. “In section 5.1, the text only talks about models m1-m6. But fig 3 also has m7. What is m7? For fig 3, please show results for all the 7 models in 3a, 3b and 3c? Why only sample 4 out of 7?”
>     - We mention m7 as necessary while answering questions 2 and 3. It is a regular model that models every possible feature in the environment (see Table 1 in the Appendix).
>     - The reason why we do not include all 7 models in the plots in Fig. 3 is that each subset of these models are useful for answering different questions. For example, in Q1 we want to compare non-VE partial models (m1, m2, m3) with minimal VE partial models (m4) one and in Q2 we want to compare minimal VE models (m4) with VE partial models (m5, m6) and a regular model (m7). The same goes for Q3 as well.
>     - The details of all these models can be found in Table 1 in the Appendix.
>
> 4. “The results are only for discrete grid-world environments, which makes it unclear if the proposed approach will generalize to more realistic settings.”
>     - We would like to note that even though we have made use of gridworlds only, we make use of them in different ways. For the squirrel’s world environment, we use a tabular agent. However, for the minigrid environments we use an agent that makes use of function approximation using neural networks. So the agent in the latter scenario is **quite general** and it can be **applied straightforwardly** to any of the popular RL environments that are used today.
>     - The specific reason for using the squirrel’s world environment was to **illustrate** the ideas in this paper and the specific use of minigrid environments was to perform **controlled** experiments in which we can test whether the agent actually only models the relevant aspects of its environment.
>     - As indicated in our conclusion, we left the part of performing more challenging experiments to future work and we are working on a follow up paper to demonstrate this part as well.
>
> 5. “While answering question 7 (on page 9), what does the search budget refer to? Planner lookahead steps or planner time cutoff or something else?”
>     - Yes, this refers to the number of lookahead steps. This part is also detailed in Algorithm 2 (see line 5) in the Appendix.
>
> 6. “The word lifelong is very misleading because it seems to indicate a continual learning flavor, which is not the case. Please consider removing the word from the title and the paper.”
>     - We would like to note that in this paper, we are targeting **lifelong learning environments** which consist of **large state spaces** and **distribution shifts** (see [1, 2]). That is why we have included the name “lifelong RL” in the title and the paper itself.
>     - We do agree that lifelong RL also refers to the scenario in which the agent faces a sequence of problems that unfold over time. However, it is just that we haven’t tackled this aspect of the problem in this study. We have made this part more clear as a limitation of our study in our Conclusion section (see the last two sentences marked in blue in the revised version of the study).
>
> [1] Tom Schaul, Hado van Hasselt, Joseph Modayil, Martha White, Adam White, Pierre-Luc Bacon, Jean Harb, Shibl Mourad, Marc Bellemare, and Doina Precup. The barbados 2018 list of open issues in continual learning. arXiv preprint arXiv:1811.07004, 2018.
>
> [2] Richard S Sutton, Michael H Bowling, and Patrick M Pilarski. The alberta plan for ai research. arXiv preprint arXiv:2208.11173, 2022.

---

> > ### Comment · Reviewer_SiWS · 2022-11-09
> > **Thank you for the clarifications**
> >
> > Thank you for providing the clarifications. I do not have further questions at the moment.

---

### Official Review · Reviewer_xzJK · 2022-11-03

**Confidence:** 4
**Correctness:** 3
**Technical Novelty And Significance:** 3
**Empirical Novelty And Significance:** 3
**Recommendation:** 5

**Clarity, Quality, Novelty And Reproducibility:**

Clarity/Quality: The work clearly states the problem, its motivation, and the proposed method. The experimental part also clearly illustrates the theoretical findings. The work is well-written and didactic.

Novelty: To the best of my knowledge, this work provides a novel contribution to the theoretical understanding of a common intuition. A “common intuition” refers to the fact that some previous empirical works already provide results in the same direction for planning/exploring in learned compact latent spaces [3] or even learning from imagined trajectories [4].

Reproducibility: On the theoretical side, all theorems are accompanied by detailed mathematical proofs and all assumptions needed. In the experimental section, all results provide confidence intervals for statistical significance. There is no mention of code being open-sourced to encourage reproducibility, nor any description of the computational costs/hardware associated with the experimental section.


**Strength And Weaknesses:**

Strengths:

- The work analyzes the problem of learning dynamics models with relevant information for planning, showing an important direction to develop principled methods. Hence, it is well-motivated.

- It formalizes the class of minimal value-equivalent partial models and provides its theoretical grounding. The definitions are clear, and the assumptions made are also clearly stated.

- The theoretical understanding provides important insights in terms of value/planning losses, as well as the considerations for the sample and computational complexity. Furthermore, the experimental methodology and empirical results support the proposed theory, raising and answering the appropriate questions with the right choice of baselines and ablations.

- Lastly, the work suggests inductive biases to scale its finding for the Deep RL scenario, and conduct ablation studies to support their recommendations.

Concerns:

- There is a limitation of the minimal VE partial models worth discussing in the paper. By definition, the relevant features are directly related to the reward and the task. Therefore, these models do not seem to be well-suited for scenarios where generalization across tasks is crucial (e.g., multi-task/meta-learning), which are important in the context of lifelong RL.

- The distribution shifts analyzed in question 6 only relate to the irrelevant aspects of the environment. It would be interesting to discuss or provide empirical results for distribution shifts in a relevant feature, or even in scenarios where the set of relevant features changes over time, especially given the motivation of lifelong RL. The hypothesis is that the suggested inductive biases are not robust enough to deal with such shifts when it needs to learn the representation.

- The current related work section does not place the work very well in the literature. For instance, I believe the work should contrast with recent literature on RL in the presence of exogenous distractors [1, 2] (once it approaches a very similar problem) and planning in learned latent spaces [3].


Further Suggestions/Minor Concerns:

- It would be interesting to provide a small high-level description of how the Straight-Forward Decision-Time Planning algorithm works, besides the pseudocode. This should help in the presentation of this baseline.

- In Question 3: “D minimal…” -> “Do minimal…”

- Last line of Section 5: “compunding” -> “compounding”



**Summary Of The Paper:**

This work introduces minimal value-equivalent partial models, which are models of the environment built on top of a minimal subset of features from the observational space and holds the property of value-equivalence property. This property ensures that the value function associated with an optimal policy obtained as a result of planning with the model is equivalent to the optimal value function in the true environment. The work theoretically analyzes the value and planning losses for such models and provides estimations for computational and sample complexity. Finally, it illustrates such insights in controlled experiments and suggests inductive biases to implement these models in the Deep RL context, with proper ablation studies.

**Summary Of The Review:**

The proposed work provides didactic definitions and theoretical grounding from planning in “partial” models. The experimental part also provided a good illustration of the insights presented, although more experiments are required to claim that the proposed inductive bias works for more complex scenarios. There are some concerns about questions and limitations to be discussed in the paper.

References

[1] Efroni et. al. Sample-Efficient Reinforcement Learning in the Presence of Exogenous Information. COLT, 2022.

[2] Efroni et. al. Provable RL with Exogenous Distractors via Multistep Inverse Dynamics. ICLR, 2022.

[3] Ekar et. al. Planning to Explore via Self-Supervised World Models. ICML, 2020.

[4] Hafner et. al. Learning Latent Dynamics for Planning from Pixels. ICML, 2019.


====================POST-REBUTTAL============================

I appreciate the authors’ efforts to clear my concerns. After considering the rebuttal and the discussion with other reviewers, I decided to change my score to {5: marginally below the acceptance threshold} for the following reason: despite having interesting theoretical contributions and experiments that illustrate them in a controlled environment, it is hard to evaluate if the proposed inductive bias (illustrated in Appendix C.5) hold for more complex scenarios. My second concern point remains open, as the paper does not provide evidence to support that this proposed inductive bias is robust to deal with more complex distribution shifts, a critical aspect of lifelong RL. This is essential to clear in the experiments once prior work often suggests that more signals are needed for effective representation learning in complex scenarios [3, 4]. As pointed out by other reviewers, this is somehow conflicting with prior literature, which makes it necessary to address for acceptance.

---

> ### Author Response · Authors · 2022-11-07
> **Regarding missing references**
>
> We would like to thank the reviewer for their detailed and constructive review. We will provide a response shortly, however, we would like to indicate that the references in this review are missing. Would the reviewer be able to add them?

---

> > ### Comment · Reviewer_xzJK · 2022-11-07
> > **Missing references**
> >
> > Thanks for pointing out the missing references. I've just added them to the official review.

---

> ### Author Response · Authors · 2022-11-07
> **Response to Reviewer xzJK (1)**
>
> We would like to thank the reviewer for their very detailed and constructive feedback on the paper. Finding reviews of this kind is rare nowadays, so we really appreciate it.
>
> **Responses to the Weaknesses part of the reviewer’s review:**
>
> 1. “There is a limitation of the minimal VE partial models worth discussing in the paper. By definition, the relevant features are directly related to the reward and the task. Therefore, these models do not seem to be well-suited for scenarios where generalization across tasks is crucial (e.g., multi-task/meta-learning), which are important in the context of lifelong RL.”
>     - We would like to thank the reviewer for bringing this up. However, we do think that minimal VE partial models **can actually be quite useful** in scenarios where generalization across tasks is crucial. For instance, consider the squirrel in the Squirrel’s World environment, most of the tasks that are of interest to the Squirrel are always related to the position of the squirrel and the position and direction of the hawk (the squirrel would never have to model e.g. the wind direction in the upper atmosphere). If the squirrel wants to navigate to the middle bushes to obtain another food or to bring the nut back to its starting position, all it has to do is to learn the **new reward function** associated with the task as it already has a good knowledge of the dynamics of the world (it does not need to relearn it).
>     - In the scenario described above, even though zero-shot generalization would not be possible, the pre-learned dynamics model would **definitely help a lot** in terms of quick adaptation to the task of interest.
>     - We are aware that this is not immediately obvious from the initial submission of the paper. Because of this, we have **updated** the conclusion section of our paper to reflect this (see the last two sentences in the Conclusion section of the revised version of the study).
>
> 2. “The distribution shifts analyzed in question 6 only relate with the irrelevant aspects of the environment. It would be interesting to discuss or provide empirical results for distribution shifts in a relevant feature, or even in scenarios where the set of relevant features changes over time, especially given the motivation of lifelong RL. The hypothesis is that the suggested inductive biases are not robust enough to deal with such shifts when it needs to learn the representation.”
>     - We would like to thank the reviewer in bringing up the topic of what will happen when the relevant aspects of the environment change. However, in this study we are focused lifelong RL scenarios in which the environment is large and **mostly contains irrelevant features** [1, 2] (much like real-life scenarios in which there are a few relevant aspects, e.g. a fish only has to model what is going on in its local area (like the behavior of its prey and other bigger fish) and does not have model all the complex fluid dynamics or the motion of the other creatures living in land).
>     - We realize that our initial submission was not particularly clear on this and thus we have **added this detail** to the Introduction section of the revised version of the study (see the first paragraph).
>     - Motivated by this, we have only considered the scenarios in which the irrelevant aspects of the agent’s environment changes over time. However, if it were the relevant aspects that were changing, we are not sure if minimal VE partial models would be helpful in that scenario. But maybe the **non-minimal ones can be helpful** as they model more than the relevant features.
>
> 3. “The current related work section does not place the work very well in the literature. For instance, I believe the work should contrast with recent literature on RL in presence of exogenous distractors [1, 2] (once it approaches a very similar problem) and planning in learned latent spaces [3].”
>     - We would like to thank the reviewer for pointing to these studies. We were not aware of them and indeed they seem to be related to our study.
>     - In the revised version of the study, we **have tried to incorporate [1, 2, 3] in our related work section**.
>     - If the reviewer has further concerns on how we relate the pointed studies to our study, we would be glad to hear about them and make the necessary corrections.
>
> 4. “Further Suggestions/Minor Concerns:”
>     - We would like to thank the reviewer for providing suggestions and pointing out our typos. The first suggestion of the reviewer would indeed be helpful in terms of improving the presentation of the baseline, however, we would like to note that we are operating on tight space constraints. That’s why we point the readers to the pseudocode.
>     - For the two typo corrections, we note that we have corrected all of them in the revised version of the study.

---

> > ### Author Response · Authors · 2022-11-07
> > **Response to Reviewer xzJK (2)**
> >
> > [1] Tom Schaul, Hado van Hasselt, Joseph Modayil, Martha White, Adam White, Pierre-Luc Bacon, Jean Harb, Shibl Mourad, Marc Bellemare, and Doina Precup. The barbados 2018 list of open issues in continual learning. arXiv preprint arXiv:1811.07004, 2018.
> >
> > [2] Richard S Sutton, Michael H Bowling, and Patrick M Pilarski. The alberta plan for ai research. arXiv preprint arXiv:2208.11173, 2022.
> >
> > **Responses to the Reproducibility part of the reviewer’s review:**
> >
> > 1. “There is no mention of code being open-sourced to encourage reproducibility, nor any description of the computational costs/hardware associated with the experimental section.”
> >     - For our experiments, we have used two different agents: (i) a tabular agent that performs value iteration, and (ii) the straightforward decision-time planning agent of Zhao et al. (2021). We thought that there is no need to provide the code for the first agent as the implementation of it is relatively straightforward. For the second agent, we have added a link to the publicly available code of Zhao et al. (2021) which we have used in this study. See page the footnote in page 16 in the revised version of the study.
> >     - Regarding the computational cost, we believe that our RL agents can be implemented with regular hardware that is available to most research groups in academia. There is no requirement for running on compute-heavy clusters.

---

> > > ### Comment · Reviewer_xzJK · 2022-11-16
> > > **Minor suggestion for reproducibility**
> > >
> > > Thank you for the clarification. I would still recommend (as a minor suggestion) adding this information regarding computational cost in the appendices to help researchers to reproduce your experiments in the future.

---

> > > > ### Author Response · Authors · 2022-11-16
> > > > **Response to minor suggestion for reproducibility**
> > > >
> > > > Sure, we will add this part to the appendix after the rebuttal period.

---

> > ### Comment · Reviewer_xzJK · 2022-11-16
> > **Thank you for the response!**
> >
> >
> > I would like to thank the authors for addressing the concerns. I am mostly clear now, with few suggestions.
> >
> > For point (1), I understand the arguments raised by the authors here. My concern was mainly in the direction where different tasks could require learning different relevant features, but I believe that the text added in the Conclusion section addressed my point.
> >
> > For point (3), I appreciate the incorporation of the related work, but it is not very clear the differences raised. For instance, Efroni et. al. also approach environments with mostly irrelevant features (namely, high dimensional, exogenous information). Furthermore, it is not clear what means “explicitly focus on the structure” of the learned representation, when contrasting with Sekar et. al.
> >
> > For the first suggestion in point (4), I understand the concerns about the space. My initial recommendation was to add a support text alongside the pseudocode in the Appendix section, which will not compromise the 9-page space from the main document. But this is a minor suggestion.

---

> > > ### Author Response · Authors · 2022-11-16
> > > **Response to Reviewer xzJK**
> > >
> > > We would like to thank the reviewer for their response.
> > >
> > > Regarding point (3), we have uploaded a revised version of the paper to clarify this part. If the reviewer still has some concerns, we would be glad to hear their advice on how we can alter this part.
> > >
> > > Regarding point (4), we will add this part to the appendix after the rebuttal period.

---

> > > > ### Author Response · Authors · 2022-11-18
> > > > **Any outstanding concerns on the related work section?**
> > > >
> > > > Dear reviewer, with our uploaded revised version of the paper, we have tried to address your concerns regarding the related work section of our paper. As there is only one day left for the discussion period to end, we would like to know if you have any outstanding concerns in light of revision? We will be happy to address them.

---

> > > > > ### Comment · Reviewer_xzJK · 2022-11-18
> > > > > **No outstanding concerns**
> > > > >
> > > > > Thanks for your message. At this moment, I do not have further concerns about the related work section.

---

> ### Author Response · Authors · 2022-11-13
> **Any outstanding concerns**
>
> Dear reviewer, with our response below, we have tried to address your concerns regarding our paper. As we are more than halfway through the discussion period, we would like to know if you have any outstanding concerns in light of our response? We will be happy to address them.

---

### Official Review · Reviewer_5G3f · 2022-11-04

**Confidence:** 4
**Correctness:** 3
**Technical Novelty And Significance:** 2
**Empirical Novelty And Significance:** 2
**Recommendation:** 5

**Clarity, Quality, Novelty And Reproducibility:**

The paper is well written with clear organization. The math is clear and easy to follow. The Appendix contains enough details to reproduce results.

**Strength And Weaknesses:**

Strengths:

The paper carries out thorough theoretical analysis of the performance gap of MVE models and their performance in terms of planning and complexity costs.

Weaknesses:

1. It is unsurprising that models which only encode task-relevant features would perform better than generic models, which might have multiple distractions. The difficulty is learning these models using salable architectures. The results presented here are incomplete and somewhat inconsistent with current model-based RL literature, i.e. value and reward signals are often insufficient to learn meaningful representations in more complex environments.
2. Related to the previous point, the experiments are limited to GridWorld domains, which are relatively straightforward not realistic. It is unclear whether the experimental results presented here would hold in larger more realistic domains (probably not, based on previous works, i.e. Dreamer/SLAC).

I am willing to increase my score if more involved experiments with realistic environments/control problems and current baselines are added.



**Summary Of The Paper:**

This work focuses on model-based RL, particular in the life-long setting. It sets up the notation of minimal value-equivalent models, which are minimal-sized models which can capture all task relevant features of the environment. The authors show theoretical results that under a particular definition of partial minimal value-equivalent models, the agent does not incur a performance cost of planning using the model, but experiences improvement in computational and sample complexity. This is verified in experiments based on the MiniGrid domain. The second part of the experimental section is concerned with learning such models from interaction with deep learning architectures. The authors experimentally find that:

1. Models which use representations trained through the loss of the value function perform and scale better than models trained through dynamics prediction as well.
2. Models trained on a variety of environments perform better than models trained on a single environment in terms of robust transfer.
3.. Models based using representations based on value-function training are more robust to model compounding errors.

**Summary Of The Review:**

The paper provides good theoretical analysis of model-based learning in life-long/multi-task setting, however these seem of limited applicability in more complex and realistic domains.

---

> ### Author Response · Authors · 2022-11-07
> **Response to Reviewer 5G3F (1)**
>
> We would like to thank the reviewer for their feedback on the paper.
>
> **Responses to the Weaknesses part of the reviewer’s review:**
>
> 1. “It is unsurprising that models which only encode task-relevant features would perform better than generic models, which might have multiple distractions. The difficulty is learning these models using salable architectures. The results presented here are incomplete and somewhat inconsistent with current model-based RL literature, i.e. value and reward signals are often insufficient to learn meaningful representations in more complex environments.”
>     - We are not sure if a paper should be judged on whether it contains surprising results or not. Afterall this is something **subjective**. The main idea of this paper was to introduce certain types of models to model-based RL to allow for performing scalable and robust planning in lifelong RL scenarios.
>     - We agree that the difficulty is in learning these models and we have **actually tried to address this problem in question 6** of the paper by providing two simple and useful heuristics in learning minimal VE partial models with **scalable** architectures like deep learning architectures. We have also performed experiments to verify whether if these heuristics work or not and showed that they can actually lead to models that display the behavior of minimal VE partial models.
>     - We are not sure what **incomplete means**? Would the reviewer be able to elaborate on that? If it is meant that this paper does not introduce all there is to planning in lifelong RL, we agree with the reviewer and note that we have **explicitly** mentioned this as a limitation of our work in the Conclusion section.
>     - We are not sure what the reviewer is referring to with our paper being inconsistent with the current model-based RL literature. Can the reviewer point to specific papers? As far as we are aware, the value and reward signals are **quite sufficient** to learn models and representations that allow for very good performance in complex environments. For example, see **MuZero** [1] (performs really well on board games and Atari) which falls under the category of value-equivalent models [2].
>
> 2. “Related to the previous point, the experiments are limited to GridWorld domains, which are relatively straightforward not realistic. It is unclear whether the experimental results presented here would hold in larger more realistic domains (probably not, based on previous works, i.e. Dreamer/SLAC).”
>     - We would like to note that even though we have made use of gridworlds only, we make use of them in different ways. For the squirrel’s world environment, we use a tabular agent. However, for the minigrid environments we use an agent that makes use of function approximation using neural networks. So the agent in the latter scenario is **quite general** and it can be **applied straightforwardly** to any of the popular RL environments that are used today.
>     - The specific reason for using the squirrel’s world environment was to **illustrate** the ideas in this paper and the specific use of minigrid environments was to perform **controlled** experiments in which we can test whether the agent actually only models the relevant aspects of its environment.
>     - As indicated in our conclusion, we left the part of performing more challenging experiments to future work and we are working on a follow up paper to demonstrate this part as well.
>     - We do **not** agree with the statement that based on previous work such as Dreamer/SLAC our results would not scale to more challenging environments. First of all, these model-based RL algorithms only consider regular benchmarks like Atari whereas the main motivation of this paper is on **lifelong learning environments** where there are distribution shifts and most of the things in the environment do not actually matter. Second, results in the value equivalence literature are actually telling that value-equivalent models can **achieve very good performances** in today’s complex RL benchmarks (see [1]).
>
>
> [1] Julian Schrittwieser, Ioannis Antonoglou, Thomas Hubert, Karen Simonyan, Laurent Sifre, Simon Schmitt, Arthur Guez, Edward Lockhart, Demis Hassabis, Thore Graepel, et al. Mastering atari, go, chess and shogi by planning with a learned model. Nature, 588(7839):604–609, 2020.
>
> [2] Christopher Grimm, Andre Barreto, Satinder Singh, and David Silver. The value equivalence principle for model-based reinforcement learning. In H. Larochelle, M. Ranzato, R. Hadsell, M.F. Balcan, and H. Lin (eds.), Advances in Neural Information Processing Systems, volume 33, pp. 5541–5552. Curran Associates, Inc., 2020. URL https://proceedings.neurips.cc/ paper/2020/file/3bb585ea00014b0e3ebe4c6dd165a358-Paper.pdf.

---

> > ### Author Response · Authors · 2022-11-07
> > **Response to Reviewer 5G3F (2)**
> >
> > 3. “I am willing to increase my score if more involved experiments with realistic environments/control problems and current baselines are added.”
> >     - We would like to note that this is not possible as we are already **out of space** and we have already filled up our experimental results section with **illustrative and controlled experiments** that are more related to the main motivation of this paper: introduce certain types of models that are relevant to performing planning in lifelong RL scenarios.
> >     - However, we do think that this is a nice future work direction and we would like to note that are working on it.

---

> > > ### Comment · Reviewer_5G3f · 2022-11-17
> > > **Thank you for your response**
> > >
> > > Thank you for your response and apologies for the late reply.
> > >
> > > 1, My overall comments of incompleteness address the experimental evaluations and baselines as discussed within this paper. The provided experiments read like a proof of concept, rather than a complete work. It is a stretch that methods which work on small gridworlds will generalize well to complex and realistic environments. Moreover most board games, have little to no irrelevant features and have pretty straightforward dynamics.
> > >
> > >
> > > 2.There exist a variety of evaluation environments and papers, which focus on extracting control-relevant features, both in the single and multi-task domain. For a reference consider [1] and the related work section (this work seems inspired by the same principles considered here) (the reviewer is not affiliated with the authors). An example is the robodesk environment with distractions, which fits nicely within the framework presented here. It has complex dynamics, which cannot be explicitly written, with multiple related tasks and rich task irrelevant distractions. It is also pretty straight-forward to construct such environments from existing meta-learning benchmarks, such as MetaWorld. Also there are a variety of more-realistic maze-like environments, i.e. DMLab, which could also fit your setting.
> > >
> > >
> > > 3. I would like to clarify my earlier statement, that reward and value signals are not enough to learn robust representations. What I mean is that this is not enough to efficiently learn such representations, consider the standard reference of Dreamer [2] (and related works section), Appendix E, which shows that models trained reward prediction fail to make timely learning progress.
> > >
> > > 4. The paper can definitely be re-organized to fit additional experiments. Presenting the developed theory with a complete set of experiments and baselines would make for a much stronger submission, rather than splitting it into two.
> > >
> > > Based on these points, I will maintain my score.
> > >
> > >
> > > [1] Denoised MDPs: Learning World Models Better Than the World Itself: Denoised MDPs: Learning World Models Better Than the World Itself.
> > >
> > > [2] Dream to Control: Learning Behaviors by Latent Imagination: Danijar Hafner, Timothy Lillicrap, Jimmy Ba, Mohammad Norouzi

---

> > > > ### Author Response · Authors · 2022-11-18
> > > > **Response to the Reviewer's response**
> > > >
> > > > We would like to thank the reviewer for their response.
> > > >
> > > > 1. "My overall comments of incompleteness address the experimental evaluations and baselines as discussed within this paper. The provided experiments read like a proof of concept, rather than a complete work."
> > > >
> > > >     - Yes, it is indeed the case that this paper was meant to introduce minimal partial VE models to the planning in lifelong RL literature. That is why the paper contains experiments that are specifically focused on showing the potential of these types of models.
> > > >     - **We are not sure if this makes the paper incomplete though**. There are certainly papers in the literature that introduce an idea and show its potential.
> > > >
> > > > 2. " It is a stretch that methods which work on small gridworlds will generalize well to complex and realistic environments. Moreover most board games, have little to no irrelevant features and have pretty straightforward dynamics."
> > > >
> > > >     - We would like to note that even though we have used gridworlds (Minigrid environments) in our function approximation experiments, the agent here takes as input the **image** of the top-down view of the grid as input. So, if we were to use an pixel-based game with the same structure (there is again irrelevant features in the environment, like a irrelevant objects flying at the top of the screen in space-invaders), we would be obtaining the **same results**. In this sense, we believe that our results **can generalize to more complex environments**.
> > > >     - We are not sure how to address the board games part as we have **not used any board games** in the paper.
> > > >
> > > > 3. "There exist a variety of evaluation environments and papers, which focus on extracting control-relevant features, both in the single and multi-task domain. For a reference consider [1] and the related work section (this work seems inspired by the same principles considered here) (the reviewer is not affiliated with the authors). An example is the robodesk environment with distractions, which fits nicely within the framework presented here. It has complex dynamics, which cannot be explicitly written, with multiple related tasks and rich task irrelevant distractions. It is also pretty straight-forward to construct such environments from existing meta-learning benchmarks, such as MetaWorld. Also there are a variety of more-realistic maze-like environments, i.e. DMLab, which could also fit your setting."
> > > >
> > > >     - We would like to thank the reviewer for pointing out to these environments. However, as stated previously, our function approximation experiments were meant to be **controlled experiments** in which we can **actually control** the environment so as to test whether if the agent learns the types of models we are interested in. And the Minigrid environments (that also allow for image inputs to the agent) seemed to be the best environment for the controlled experiments.
> > > >     - We will definitely check out the provided reference for future work on partial VE models.
> > > >
> > > > 4. "I would like to clarify my earlier statement, that reward and value signals are not enough to learn robust representations. What I mean is that this is not enough to efficiently learn such representations, consider the standard reference of Dreamer [2] (and related works section), Appendix E, which shows that models trained reward prediction fail to make timely learning progress."
> > > >
> > > >     - We would like to note that Dreamer is explicitly about performing model-based RL in benchmarks where **the agent is tested on the environments that it was trained on**. In this sense, it is of course expected from the agent's model to benefit from learning the dynamics in addition to the reward and value.
> > > >     - In our paper, we are interested in learning models that are **useful for performing transfer in which the test and training environments are different**. In this sense, **we would expect the results from the Dreamer paper to not generalize to our setting**.
> > > >
> > > > 5. "The paper can definitely be re-organized to fit additional experiments. Presenting the developed theory with a complete set of experiments and baselines would make for a much stronger submission, rather than splitting it into two."
> > > >
> > > >     - We are not sure how to address this concern on how to fit additional experiments to the paper (with the 9 page limit) as we are not sure how we can add additional experiments without removing the ones that already exists.
> > > >     - That being said, we would like to note that we believe that our experiments serve well in introducing the potential of minimal partial VE models and in showing how they can be learned with function approximation. As a study whose **main aim is to introduce these types of models to the lifelong RL literature**, we believe that our experiments are sufficient.

---

> ### Author Response · Authors · 2022-11-13
> **Any outstanding concerns**
>
> Dear reviewer, with our response below, we have tried to address your concerns regarding our paper. As we are more than halfway through the discussion period, we would like to know if you have any outstanding concerns in light of our response? We will be happy to address them.

---

### Author Response · Authors · 2022-11-07
**Revised Version**

Dear reviewers, we have uploaded a revised version of our paper to reflect your comments. For your convenience, most of the changes are marked in blue color.

---

### Author Response · Authors · 2022-11-19
**Less than 12 hours left for the discussion period to end**

Dear reviewers 5G3f and oVRM, with our responses below, we have tried to address your concerns regarding our paper. As there is only less than **12 hours** left for the discussion period to end, we would like to know if you have any outstanding concerns in light of our response? We will be happy to address them.

---

### Decision · Program_Chairs · 2023-01-20

**Decision:**

Reject

**Justification For Why Not Higher Score:**

The effectiveness of the propose approach beyond grid world, particularly in the lifelong learning scenarios (which is the main motivation of the work), is questionable.

**Justification For Why Not Lower Score:**

N/A

**Metareview: Summary, Strengths And Weaknesses:**

This paper proposes a model-based reinforcement learning (RL) method with a novel notion of minimal value-equivalent model, which encodes task-relevant features and improves computational and sample complexity.  The main strength of the paper is in its theoretical analysis that shows the losses in value and planning as well as the computational and sample complexity in using the minimal value-equivalent model.

While the proposed approach is validated with grid world tasks, it is not quite not convincing that if the proposed approach work effectively in domains more complex domains, particularly in the scenarios of lifelong learning, contrary to the claim of the paper.